# Glimpse: Geometry Learning of Multi-scale Structural Priors for 3D Pose Estimation

**Zhenhua Tang** [1]  **Jihua Peng** [2]  **Yanbin Hao** [3]  **Qiguang Miao** [4]  **Chi-Man Pun** [1]

## Abstract

Monocular 3D human pose estimation is fundamentally challenged by severe occlusion and inherent depth ambiguity. To address this, we propose Glimpse, a framework that learns robust 3D poses by explicitly modeling anatomical geometry from a single image. We recast the problem as geometry learning of multi-scale structural priors, realized through two synergistic components. First, structured sampling captures the body's geometric continuity through dual-level feature extraction, acquiring both local joint appearance and continuous features along skeletal limbs via deformable sampling. By propagating limb-level geometric cues to their connected joints, this design bridges information gaps caused by occlusion. Second, geometric correction ensures global 3D consistency by lifting coherent 2D features into a canonical 3D reference space, where a shared 3D anchor guides a distance-aware fusion mechanism. Extensive experiments conducted on Human3.6M and MPI-INF-3DHP demonstrate that Glimpse achieves state-of-the-art performance, with superior robustness under severe occlusion and complex articulation. Code is available at https://github.com/zhenhuat/Glimpse.git.

## 1. Introduction

Monocular 3D human pose estimation (3D-HPE) is fundamental yet challenging due to inherent depth ambiguity and frequent occlusions. The prevailing two-stage

---
[1]Faculty of Science and Technology, University of Macau, Macau SAR, China. [2]Division of Integrative Systems and Design, Hong Kong University of Science and Technology, Hong Kong SAR, China. [3]School of Computer and Information Engineering, Hefei University of Technology, Anhui, China. [4]School of Computer Science and Technology, Xidian University, Shaanxi, China. Correspondence to: Chi-Man Pun <cmpun@umac.mo; zhenhuat@foxmail.com>.

*Proceedings of the 43rd International Conference on Machine Learning*, Seoul, South Korea. PMLR 306, 2026. Copyright 2026 by the author(s).

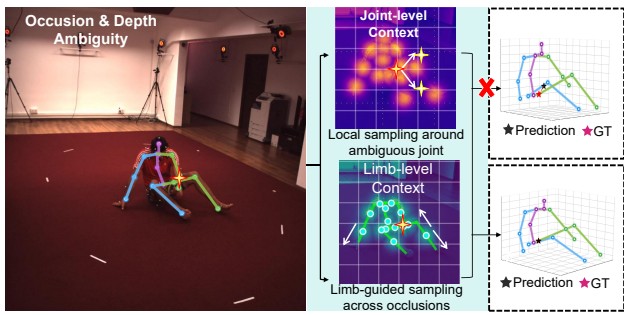

*Figure 1.* Illustration of the structured sampling strategy. Our method extracts both joint-level appearance features and limb-level geometric continuity, enabling robust 3D human pose estimation under self-occlusion.

paradigm (Martinez et al., 2017), which lifts detected 2D keypoints to 3D, becomes fragile under these conditions, as unreliable 2D detections often lead to anatomically implausible 3D reconstructions. While leveraging temporal cues across video sequences can alleviate ambiguity (Hossain & Little, 2018; Pavllo et al., 2019; Liu et al., 2020), it incurs high computational overhead and latency, making it unsuitable for real-time applications.

Within the single-frame setting, numerous methods (Zhao et al., 2019; Xu & Takano, 2021; Zeng et al., 2021; Zhao et al., 2022; Tang et al., 2023b; Li et al., 2025b) rely solely on sparse 2D joint coordinates and adopt varied architectural designs to model dependencies among joints. However, by decoupling structural reasoning from visual content, these approaches become vulnerable to inaccuracies or detection failures from upstream 2D pose estimators (Chen et al., 2018; Sun et al., 2019), particularly under occlusion. Recent efforts (Zhao et al., 2023; Zhou et al., 2024; Zheng et al., 2026) address this issue by incorporating multi-scale image features, yet they typically sample image features only at isolated 2D joint locations to model visual context. While this strategy enhances robustness by providing local appearance cues, it essentially treats the human body as a collection of spatially fragmented feature points. Consequently, this paradigm limits the model's ability to leverage the rich, continuous structural information inherent in multi-scale image feature maps, such as coherence along skeletal limbs.

To overcome these limitations, we revisit the design of contextual modeling in 3D-HPE. Rather than aggregating features from isolated joints, we posit that explicitly learning a continuous geometric prior from visual features is key to robust and accurate estimation. In this paper, we propose Glimpse, a framework that achieves **G**eometry **L**earning of **M**ulti-scale Structural **P**riors for 3D Pose **E**stimation. Glimpse introduces two synergistic components: structured sampling and geometric correction.

Structured sampling explicitly models the body's geometric structure in the image plane. It leverages the insight that visible limb segments provide strong geometric constraints for inferring connected joints, even under occlusion. Guided by a skeletal graph, this module extracts multi-scale features through a dual-path design. At the joint level, it captures local appearance via deformable sampling; at the limb level, it performs continuous feature sampling along and across each skeletal segment. This enables the model to explicitly encode structural continuity along skeletal directions and mid-limb visual context, information that is missing from isolated joint features. The resulting limb-level features are then propagated back through the adjacency matrix of the skeletal graph, iteratively enhancing the feature representations of their connected joints. This allows visible limbs to supply geometric and appearance cues for inferring occluded joints, thereby bridging the information gaps caused by occlusion. Meanwhile, in both paths, deformable sampling allows the network to explore surrounding regions, with learned attention weights and channel-wise gating naturally suppressing irrelevant off-body signals.

Geometric correction then transforms this geometrically coherent 2D evidence into a globally consistent 3D pose representation. The module establishes a canonical 3D reference through a learnable anchor point in the 3D space, which is projected into the feature space to serve as a shared structural reference. For each joint, the module computes geometry-aware similarity weights between its multi-scale features and this anchor point, using a Gaussian kernel centered on the joint's 3D position relative to the anchor. This mechanism adaptively selects and integrates feature scales that are most consistent with the global reference, thereby enhancing the robustness of the representation against local ambiguities or noise. Subsequently, the selected features are nonlinearly fused with the anchor embedding through a gated fusion layer, which modulates the contribution of visual evidence relative to the canonical reference. This process ensures that the final 3D representation of each joint remains both faithful to the image content and geometrically consistent with the canonical 3D reference.

The structured sampling and geometric correction components work in synergy within Glimpse, enabling the explicit learning and enforcement of body geometry from single images. In summary, our main contributions are threefold:

- **Structured sampling** jointly captures local joint appearance and limb-wise geometric continuity. It enables robust inference under occlusion by propagating visual cues from visible to occluded joints.

- **Geometric correction** lifts and aligns 2D features into a consistent 3D reference space via distance-aware fusion. It resolves local ambiguities and ensures global pose consistency.

- **The Glimpse framework** integrates both components into an efficient model. It achieves state-of-the-art performance with minimal overhead, demonstrating strong robustness to input noise and occlusion.

## 2. Related Work

Our work addresses the challenges of monocular 3D-HPE, specifically aiming to resolve occlusion and depth ambiguity within the single-frame setting. We contextualize our approach against existing pose estimation pipelines and geometric reasoning methods.

### 2.1. Monocular 3D-HPE Pipelines

Existing monocular 3D-HPE approaches are broadly categorized into direct regression and two-stage lifting paradigms. Direct methods regress 3D poses from images in an end-to-end fashion, utilizing representations such as volumetric grids (Pavlakos et al., 2017; Sun et al., 2018) or direct coordinate prediction (Kanazawa et al., 2018; Kolotouros et al., 2019). While conceptually straightforward, these methods contend with the high non-linearity of the image-to-3D mapping. In contrast, the two-stage paradigm first detects 2D keypoints and subsequently lifts them to 3D, gaining dominance due to its efficiency and stability (Martinez et al., 2017). However, this approach relies on sparse 2D coordinates that naturally lack sufficient depth information, making the lifting process fragile to inaccuracies or failures in the 2D detections (Chen et al., 2018; Sun et al., 2019), especially under occlusion or challenging viewpoints.

To circumvent this ambiguity, substantial work leverages temporal consistency across video sequences. Architectures based on convolutional networks (Hossain & Little, 2018; Pavllo et al., 2019; Liu et al., 2020; Cai et al., 2019; Xu & Takano, 2021; Tang et al., 2023a), Transformers (Zheng et al., 2021; Li et al., 2022a;b; Hassanin et al., 2025; Xue et al., 2022; Zhang et al., 2022; Tang et al., 2023c; Peng et al., 2024; Li et al., 2025a), and state-space models (Huang et al., 2025; Lu et al., 2025) achieve accuracy by modeling spatio-temporal dependencies. Although effective, these video-based methods introduce computational overhead and latency, limiting their suitability for real-time applications.

Moreover, their performance remains contingent on accurate per-frame 2D pose estimation; errors in individual frames can propagate through the temporal model, compromising overall 3D accuracy.

## 2.2. Single-Frame Geometric Reasoning

Research within the single-frame regime has evolved from modeling joint dependencies in isolation to incorporating visual evidence. Early graph-based methods (Zhao et al., 2019; Xu & Takano, 2021; Zeng et al., 2021) employed Graph Convolutional Networks (GCNs) to encode anatomical priors via skeletal graphs. However, this paradigm remains fundamentally limited by its reliance on sparse and potentially noisy 2D inputs.

To mitigate this issue, recent works can be categorized into two paradigms. On one hand, generative models such as DiffPose (Gong et al., 2023) (diffusion-based) and HiPart (Zheng et al., 2025) (hierarchical transformer-based) learn to sample plausible 3D poses from a data-driven manifold, bypassing the need for explicit visual grounding. On the other hand, another line of work seeks to anchor predictions directly in image evidence by integrating visual features. For instance, CA-PF (Zhao et al., 2023) extracts and fuses multi-scale convolutional features centered at detected 2D joint locations; Zhou et al. (Zhou et al., 2024) introduce a pose-guided transformer that allows keypoints to attend to relevant image regions; and PandaPose (Zheng et al., 2026) augments the lifting process with a monocular depth estimation network to provide geometric priors.

Despite advances, these methods lack a coherent geometric representation. They neither capture continuous 2D limb geometry nor establish a canonical 3D reference to ensure global consistency during lifting. To address this, we propose the Glimpse framework. It advances geometric reasoning through two novel components: structured sampling and geometric correction.

## 3. Method

Our framework, **Glimpse**, learns multi-scale structural priors for robust single-image 3D human pose estimation. As illustrated in Figure 2, it comprises three stages: (1) **Input Processing** to extract image features and a 2D pose embedding, (2) **Geometry Learning** to explicitly capture the body's structural geometry via two novel components: structured sampling and geometric correction, and (3) **Feature Fusion and Regression** for final 3D prediction.

### 3.1. Input Processing

Given an RGB image $\mathbf{I} \in \mathbb{R}^{H \times W \times 3}$, we extract multi-scale feature maps $\{\mathbf{F}_s\}_{s=1}^{S}$ using a backbone network (e.g., HRNet-W32 (Sun et al., 2019)), where $S = 4$. A pre-

trained 2D pose detector (Chen et al., 2018) provides joint coordinates $\mathbf{P} = [\mathbf{p}_1, \ldots, \mathbf{p}_J] \in \mathbb{R}^{J \times 2}$ with $J = 17$.

We then encode the 2D pose into a latent representation and add sinusoidal positional encoding $\mathbf{E}_{\text{pos}}$:

$$\mathbf{X}^{\text{pose}} = \mathbf{W}_{\text{emb}}\mathbf{P} + \mathbf{b}_{\text{emb}} + \mathbf{E}_{\text{pos}} \in \mathbb{R}^{J \times d}. \quad (1)$$

For each scale $s$, we sample visual features from $\mathbf{F}_s$ and project them to dimension $d$:

$$\mathbf{X}^{(s)} = \mathbf{W}_s^{\text{feat}} \cdot \texttt{GridSample}(\mathbf{F}_s, \mathbf{P}) \in \mathbb{R}^{J \times d}, \quad (2)$$

where $\texttt{GridSample}$ performs bilinear interpolation. The initial pose embedding $\mathbf{X}^{\text{pose}}$ and the visual features $\{\mathbf{X}^{(s)}\}_{s=1}^{S}$ form the input tokens for geometric learning.

### 3.2. Geometric Learning

The geometric learning stage consists of two complementary components that work together to capture 2D geometric continuity and lift it into a globally consistent 3D pose, thereby learning the body's underlying structural geometry.

#### 3.2.1. STRUCTURED SAMPLING

The structured sampling component explicitly models the body's anatomical topology to extract multi-scale visual features that conform to biological constraints.

**Skeletal Topology** We model the human body as a graph $\mathcal{G} = (\mathcal{J}, \mathcal{L})$, where $\mathcal{J}$ is the set of $J$ joints and $\mathcal{L} \subset \mathcal{J} \times \mathcal{J}$ is the set of anatomical limbs (e.g., shoulder–elbow, elbow–wrist). This fixed topology provides the structural prior that guides our feature sampling, ensuring that the learned representations respect biological constraints.

**Query Token Construction** To guide the deformable sampling process, we construct query tokens that combine geometric and appearance information. For each joint $j$ and scale $s$, we fuse the pose embedding $\mathbf{x}_j^{\text{pose}}$ with the visual feature $\mathbf{x}_j^{(s)}$ at that scale to form a scale-specific query token:

$$\mathbf{q}_j^{(s)} = \text{MLP}_{\text{fuse}}\left(\mathbf{x}_j^{\text{pose}} + \mathbf{x}_j^{(s)}\right) \in \mathbb{R}^d, \quad (3)$$

where $\text{MLP}_{\text{fuse}}$ is a lightweight multi-layer perceptron. This query token $\mathbf{q}_j^{(s)}$ encapsulates both the geometric prior from the 2D pose and the appearance cues from the image at scale $s$, providing comprehensive guidance for feature sampling.

**Joint-level Sampling** For each joint $j$ at scale $s$, we use the query token $\mathbf{q}_j^{(s)}$ to predict $K = 4$ sampling offsets and corresponding attention weights. The sampling offsets are computed as:

$$\Delta \mathbf{p}_{j,s}^{(k)} = \tanh\left(\mathbf{W}_{\text{off}}\mathbf{q}_j^{(s)}\right) \in \mathbb{R}^2, \quad k = 1, \ldots, K. \quad (4)$$

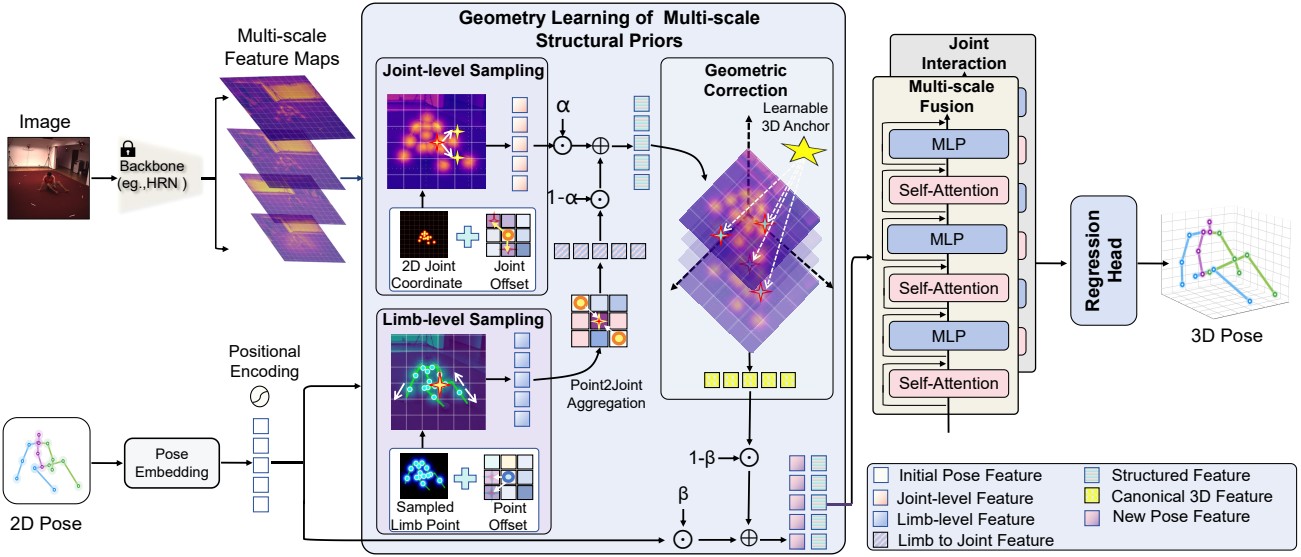

*Figure 2.* **Glimpse Framework Overview.** Our method comprises three components: (1) Input Processing via a backbone and 2D pose estimator; (2) Geometry Learning, which explicitly models anatomical structure through structured sampling and geometric correction; (3) Feature Fusion and Regression for final 3D pose estimation.

The refined sampling locations are $\hat{\mathbf{p}}_{j,s}^{(k)} = \mathbf{p}_j + \Delta \mathbf{p}_{j,s}^{(k)}$. Attention weights are computed via softmax over a linear projection of $\mathbf{q}_j^{(s)}$:

$$w_{j,s}^{(k)} = \frac{\exp \left( \mathbf{w}_{att} \mathbf{q}_j^{(s)} \right)}{\sum_{k'=1}^{K} \exp \left( \mathbf{w}_{att} \mathbf{q}_j^{(s)} \right)}. \qquad (5)$$

Features are sampled from the backbone feature map $\mathbf{F}_s$ at these locations:

$$\mathbf{v}_{j,s}^{(k)} = \texttt{GridSample} \left( \mathbf{F}_s, \hat{\mathbf{p}}_{j,s}^{(k)} \right) \in \mathbb{R}^d. \qquad (6)$$

The joint-level feature at scale $s$ is then aggregated as:

$$\mathbf{h}_j^{\text{joint},(s)} = \sum_{k=1}^{K} w_{j,s}^{(k)} \cdot \mathbf{v}_{j,s}^{(k)} \in \mathbb{R}^d. \qquad (7)$$

**Limb-level Sampling** To capture continuous limb geometry, we define a set of anatomical limb pairs $\mathcal{L}$. For each limb connecting joints $a$ and $b$, denoted as $(a,b) \in \mathcal{L}$ (e.g., shoulder–elbow, elbow–wrist, etc.), we sample $M = 3$ intermediate points along the line segment between joints $a$ and $b$:

$$\begin{aligned} \mathbf{q}_m^{(a,b)} &= (1 - \tau_m)\mathbf{p}_a + \tau_m \mathbf{p}_b, \\ \tau_m &= \frac{m}{M+1}, \quad m = 1, \dots, M. \end{aligned} \qquad (8)$$

For each limb point at scale $s$, we compute its query token as the average of the endpoint joint query tokens at that scale:

$$\mathbf{q}_{m,(a,b)}^{(s)} = \frac{1}{2} \left( \mathbf{q}_a^{(s)} + \mathbf{q}_b^{(s)} \right) \in \mathbb{R}^d. \qquad (9)$$

Using this limb query token, we predict $K = 4$ sampling offsets $\Delta \mathbf{q}_{m,s}^{(a,b,k)}$ and corresponding attention weights $\omega_{m,s}^{(a,b,k)}$ for each point:

$$\Delta \mathbf{q}_{m,s}^{(a,b,k)} = \tanh \left( \mathbf{W}_{\text{off}}^{\text{limb}} \mathbf{q}_{m,(a,b)}^{(s)} \right) \in \mathbb{R}^2, \quad k = 1, \dots, K, \qquad (10)$$

$$\omega_{m,s}^{(a,b,k)} = \frac{\exp \left( \mathbf{w}_{att}^{\text{limb}} \mathbf{q}_{m,(a,b)}^{(s)} \right)}{\sum_{k'=1}^{K} \exp \left( \mathbf{w}_{att}^{\text{limb}} \mathbf{q}_{m,(a,b)}^{(s)} \right)}. \qquad (11)$$

The refined sampling locations are $\hat{\mathbf{q}}_{m,s}^{(a,b,k)} = \mathbf{q}_m^{(a,b)} + \Delta \mathbf{q}_{m,s}^{(a,b,k)}$. Features are sampled and aggregated:

$$\mathbf{v}_{m,s}^{(a,b,k)} = \texttt{GridSample} \left( \mathbf{F}_s, \hat{\mathbf{q}}_{m,s}^{(a,b,k)} \right) \in \mathbb{R}^d, \qquad (12)$$

$$\mathbf{h}_m^{(a,b),(s)} = \sum_{k=1}^{K} \omega_{m,s}^{(a,b,k)} \cdot \mathbf{v}_{m,s}^{(a,b,k)} \in \mathbb{R}^d. \qquad (13)$$

**Limb-to-Joint Feature Fusion** For each joint $j$, we gather limb-level features from all limbs $(a,b) \in \mathcal{L}$ where $j = a$ or $j = b$, and compute the aggregated limb feature $\mathbf{h}_j^{\text{limb},(s)}$ as their average. The final structured feature for joint $j$ at scale $s$ combines both joint-level appearance and limb-level geometric information through a channel-wise gating mechanism. We introduce a learnable gating vector $\boldsymbol{\alpha} \in \mathbb{R}^d$ that applies element-wise modulation:

$$\mathbf{h}_j^{(s)} = \boldsymbol{\alpha} \odot \mathbf{h}_j^{\text{joint},(s)} + (1 - \boldsymbol{\alpha}) \odot \mathbf{h}_j^{\text{limb},(s)} \in \mathbb{R}^d, \qquad (14)$$

where $\odot$ denotes element-wise multiplication. The gating vector $\boldsymbol{\alpha}$ is obtained by applying the sigmoid function $\sigma(\cdot)$

to a learnable parameter vector $\mathbf{p}_\alpha \in \mathbb{R}^d$, i.e., $\boldsymbol{\alpha} = \sigma(\mathbf{p}_\alpha)$. This ensures each value in $\boldsymbol{\alpha}$ lies in $(0, 1)$, allowing each feature channel to independently balance the contributions from joint appearance and limb geometry.

### 3.2.2. GEOMETRIC CORRECTION

The geometric correction component aligns the multi-scale structured features with a canonical 3D structural prior to enforce global consistency.

**Canonical Reference Initialization** We introduce a learnable 3D anchor point $\mathbf{a} \in \mathbb{R}^3$ that serves as a shared structural prior. It is projected into the feature space to obtain the canonical reference:

$$\mathbf{r} = \text{MLP}_{\text{embed}}(\mathbf{a}) \in \mathbb{R}^d. \qquad (15)$$

**Distance-Aware Feature Alignment** For each joint $j$, we compute alignment weights between its multi-scale features $\{\mathbf{h}_j^{(s)}\}_{s=1}^S$ and the reference $\mathbf{r}$ via a distance-aware Gaussian kernel:

$$w_{j,s} = \frac{\exp\left(-\|\mathbf{h}_j^{(s)} - \mathbf{r}\|^2/(2\sigma^2)\right)}{\sum_{s'=1}^S \exp\left(-\|\mathbf{h}_j^{(s')} - \mathbf{r}\|^2/(2\sigma^2)\right)}, \qquad (16)$$

where $\sigma = 1.0$ is the scale parameter of the Gaussian kernel. Although $\mathbf{r}$ is shared, the weights $w_{j,s}$ are joint-specific due to the distinct local information in each $\mathbf{h}_j^{(s)}$, thereby promoting geometric consistency relative to the canonical reference.

Each feature is first enriched by incorporating the global structural prior through a nonlinear fusion with the reference $\mathbf{r}$:

$$\tilde{\mathbf{h}}_j^{(s)} = \text{MLP}_{\text{fuse}}\left([\mathbf{r}; \mathbf{h}_j^{(s)}]\right) \in \mathbb{R}^d, \qquad (17)$$

where $[;]$ denotes concatenation. Subsequently, the geometry-corrected feature for joint $j$ is obtained by aggregating these enriched features according to the alignment weights:

$$\mathbf{c}_j = \sum_{s=1}^S w_{j,s} \cdot \tilde{\mathbf{h}}_j^{(s)} \in \mathbb{R}^d. \qquad (18)$$

**Pose Embedding Update** Finally, the pose embedding is updated by modulating the geometry-corrected feature through channel-wise gating:

$$\mathbf{x}_{\text{new},j}^{\text{pose}} = \boldsymbol{\beta} \odot \mathbf{x}_j^{\text{pose}} + (1 - \boldsymbol{\beta}) \odot \mathbf{c}_j, \qquad (19)$$

where $\boldsymbol{\beta} = \sigma(\mathbf{p}_\beta) \in (0, 1)^d$ is a gating vector obtained from a learnable parameter $\mathbf{p}_\beta \in \mathbb{R}^d$.

### 3.3. Feature Fusion and Regression

The updated pose prior $\mathbf{X}_{\text{new}}^{\text{pose}} = [\mathbf{x}_{\text{new},1}^{\text{pose}}, \ldots, \mathbf{x}_{\text{new},J}^{\text{pose}}] \in \mathbb{R}^{J \times d}$ and the structured features $\{\mathbf{H}^{(s)}\}_{s=1}^S$ with $\mathbf{H}^{(s)} = [\mathbf{h}_1^{(s)}, \ldots, \mathbf{h}_J^{(s)}] \in \mathbb{R}^{J \times d}$ are concatenated into a joint-scale token sequence:

$$\mathbf{X} = [\mathbf{X}_{\text{new}}^{\text{pose}}; \mathbf{H}^{(1)}, \ldots, \mathbf{H}^{(S)}] \in \mathbb{R}^{(1+S) \times J \times d}. \qquad (20)$$

The sequence $\mathbf{X}$ is then refined through two Transformer-based stages. First, a multi-scale fusion Transformer encoder models interactions among different scales within each joint. The outputs for each joint are concatenated into a joint-wise representation. Second, a joint interaction Transformer encoder models anatomical dependencies across all joints using these representations, producing the final feature $\mathbf{X}_{\text{final}} \in \mathbb{R}^{J \times d}$.

Finally, $\mathbf{X}_{\text{final}}$ is regressed to 3D coordinates via a linear projection:

$$\hat{\mathbf{P}}^{\text{3D}} = \mathbf{W}_{\text{reg}}\mathbf{X}_{\text{final}} + \mathbf{b}_{\text{reg}}, \qquad (21)$$

where $\hat{\mathbf{P}}^{\text{3D}} = [\hat{\mathbf{p}}_1^{\text{3D}}, \ldots, \hat{\mathbf{p}}_J^{\text{3D}}] \in \mathbb{R}^{J \times 3}$ contains the predicted 3D joint positions. The model is trained end-to-end using the Mean Per Joint Position Error (MPJPE) loss:

$$\mathcal{L} = \frac{1}{J} \sum_{j=1}^J \|\hat{\mathbf{p}}_j^{\text{3D}} - \mathbf{p}_j^{\text{gt}}\|_2, \qquad (22)$$

where $\mathbf{p}_j^{\text{gt}} \in \mathbb{R}^3$ is the ground-truth 3D position of joint $j$.

## 4. Experiments

We comprehensively evaluate the proposed Glimpse framework on two standard benchmarks, Human3.6M (Ionescu et al., 2013) and MPI-INF-3DHP (Mehta et al., 2017).

### 4.1. Datasets and Evaluation Metrics

**Human3.6M** (Ionescu et al., 2013) is the most widely used indoor benchmark for 3D-HPE, containing 3.6 million video frames from 11 subjects performing 15 actions. Following the standard protocol, we use subjects 1, 5, 6, 7, and 8 for training, and subjects 9 and 11 for evaluation. We report the Mean Per Joint Position Error (MPJPE) and Procrustes-Aligned MPJPE (PA-MPJPE) in millimeters.

**MPI-INF-3DHP** (Mehta et al., 2017) and **3DPW** (Von Marcard et al., 2018) are challenging in-the-wild datasets featuring diverse activities and outdoor scenes. For MPI-INF-3DHP, following prior work (Zheng et al., 2021; Chen et al., 2021; Shan et al., 2022), we adopt MPJPE, PCK (threshold 150mm), and AUC as evaluation metrics. For 3DPW, we report MPJPE and PA-MPJPE for cross-dataset evaluation.

*Table 1.* Quantitative comparison with state-of-the-art methods on Human3.6M. MPJPE and PA-MPJPE are reported in millimeters. The suffix -HR32/-HR48 denotes the backbone width (e.g., HRNet-W32 and HRNet-W48). The best results are shown in **bold**.

| Category | Method | Publication | Frame | Parameters (M) | MPJPE↓ | PA-MPJPE↓ |
|---|---|---|---|---|---|---|
| Multi-frame | PoseFormer (Zheng et al., 2021) | ICCV'21 | 81 | 9.5 | 44.3 | 34.6 |
| | MHFormer (Li et al., 2022b) | CVPR'22 | 351 | 24.8 | 43.0 | 34.4 |
| | MixSTE (Zhang et al., 2022) | CVPR'22 | 243 | 33.6 | 40.9 | 32.6 |
| | P-STMO (Shan et al., 2022) | ECCV'22 | 243 | 4.6 | 43.0 | 34.4 |
| | STCFormer (Tang et al., 2023c) | CVPR'23 | 243 | 18.9 | 41.0 | 32.0 |
| | KTPFormer (Peng et al., 2024) | CVPR'24 | 243 | 35.2 | 40.1 | 31.9 |
| Single frame | GraphSH (Xu & Takano, 2021) | CVPR'21 | 1 | 3.7 | 51.9 | – |
| | HCSF (Zeng et al., 2021) | ICCV'21 | 1 | – | 47.9 | 39.0 |
| | GraFormer (Zhao et al., 2022) | CVPR'22 | 1 | – | 51.8 | – |
| | Diffpose (Gong et al., 2023) | CVPR'23 | 1 | 1.9 | 49.7 | – |
| | GraphMLP (Li et al., 2025b) | PR'25 | 1 | 9.49 | 49.2 | – |
| | Lifting (Zhou et al., 2024) | AAAI'24 | 1 | – | 46.4 | – |
| | HiPART (Zheng et al., 2025) | CVPR'25 | 1 | 2.4 | 42.0 | – |
| | CA-PF-HR32 (Zhao et al., 2023) | NeurIPS'23 | 1 | 14.1 | 41.4 | 33.5 |
| | CA-PF-HR48 (Zhao et al., 2023) | NeurIPS'23 | 1 | 14.1 | 39.8 | 32.7 |
| | PandaPose (Zheng et al., 2026) | NeurIPS'25 | 1 | 15.3 | 39.8 | 32.7 |
| | **Glimpse-HR32 (Ours)** | | 1 | 14.3 | **39.5** (0.3↓) | **32.5** (0.2↓) |
| | **Glimpse-HR48 (Ours)** | | 1 | 14.3 | **39.3** (0.5↓) | **32.3** (0.4↓) |

## 4.2. Implementation Details

We implement Glimpse in PyTorch and train it on a single NVIDIA RTX A6000 GPU. Following common practice (Zhao et al., 2023; Zheng et al., 2026), we use a frozen, pre-trained HRNet (Sun et al., 2019) as the backbone for multi-scale feature extraction. For fair comparison with recent single-frame methods, all experiments use detected 2D poses from CPN (Chen et al., 2018) unless otherwise specified. We train for 60 epochs with a batch size of 128, using the Adam optimizer with an initial learning rate of $1 \times 10^{-3}$. The unified feature dimension $d$ is set to 128. The feature fusion module comprises 4 transformer encoder layers with 8 attention heads.

## 4.3. Performance Comparison on Human3.6M

Table 1 compares Glimpse with state-of-the-art methods on Human3.6M. Our Glimpse-HR32 achieves an MPJPE of 39.5 mm and a PA-MPJPE of 32.5 mm. When equipped with a more powerful backbone (HRNet-W48), the Glimpse-HR48 variant achieves an even lower MPJPE of 39.3 mm, establishing a new state-of-the-art among single-frame methods and demonstrating several key advantages. First, compared to sequence-based models that require long temporal windows, such as MixSTE (Zhang et al., 2022) (40.9 mm, 243 frames) and KTPFormer (Peng et al., 2024) (40.1 mm, 243 frames), Glimpse achieves superior or comparable accuracy using only a single frame and with significantly fewer

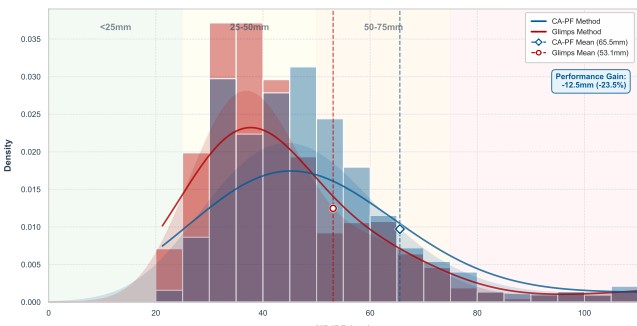

*Figure 3.* MPJPE distribution comparison on an occluded sequence. The kernel density estimate and histogram contrast CA-PF-HR32 (Zhao et al., 2023) and Glimpse-HR32 on the Sitting Down action performed by Subject S9.

parameters (14.3M vs. 35.2M). Second, compared to recent single-frame methods that operate solely on 2D coordinates, such as GraphMLP (Li et al., 2025b) and HiPART (Zheng et al., 2025) , Glimpse achieves a reduction of over 2 mm in MPJPE, validating the necessity of integrating visual evidence. Most importantly, Glimpse substantially outperforms recent context-aware single-frame approaches, including Lifting (Zhou et al., 2024) (46.4 mm), CA-PF-HR32 (Zhao et al., 2023) (41.4 mm), and PandaPose (Zheng et al., 2026) (39.8 mm), while using fewer parameters than PandaPose (14.3M vs. 15.3M). These consistent gains confirm that the explicit geometric learning paradigm is highly effective for 3D-HPE.

*Table 2.* Quantitative comparison with state-of-the-art methods on MPI-INF-3DHP. The best results are shown in **bold**.

| Method | Frame | PCK ↑ | AUC ↑ | MPJPE ↓ |
|---|---|---|---|---|
| *Multi-frame* | | | | |
| PoseFormer (Zheng et al., 2021) | 9 | 88.6 | 56.4 | 77.1 |
| MHFormer (Li et al., 2022b) | 9 | 93.8 | 63.3 | 58.0 |
| MixSTE (Zhang et al., 2022) | 27 | 94.4 | 66.5 | 54.9 |
| P-STMO (Shan et al., 2022) | 81 | 97.9 | 75.8 | 32.2 |
| D3DP (Shan et al., 2023) | 243 | 97.7 | 78.2 | 29.7 |
| *Single Frame* | | | | |
| HCSF (Zeng et al., 2021) | 1 | 82.1 | 46.2 | – |
| POT (Li et al., 2023) | 1 | 84.1 | 53.7 | – |
| Lifting (Zhou et al., 2024) | 1 | 88.2 | 59.3 | 68.9 |
| CA-PF-HR32 (Zhao et al., 2023) | 1 | 98.0 | 75.4 | 32.7 |
| CA-PF-HR48 (Zhao et al., 2023) | 1 | 98.2 | 76.3 | 31.4 |
| PandaPose (Zheng et al., 2026) | 1 | 98.6 | 75.8 | 31.8 |
| **Glimpse-HR32 (Ours)** | 1 | **98.7** (0.1↑) | 80.3(4.0↑) | **30.2** (1.2↓) |
| **Glimpse-HR48 (Ours)** | 1 | **98.7** (0.1↑) | 80.9(4.6↑) | **29.4** (2.0↓) |

*Table 3.* Cross-Dataset Generalization on 3DPW.

| Method | MPJPE | PA-MPJPE |
|---|---|---|
| CA-PF-HR32 | 138.9 | 90.1 |
| Glimpse-HR32 | **132.2** | **89.0** |

To evaluate robustness under occlusion, we conduct a targeted analysis on the challenging "Sitting Down" action from subject S9 of Human3.6M. This sequence features severe self-occlusion, particularly in limb-crossing scenarios. As illustrated in Figure 3, our Glimpse-HR32 achieves an MPJPE of 53.1 mm across 2932 samples, representing a substantial improvement of 12.4 mm (18.9%) over the CA-PF-HR32 baseline (Zhao et al., 2023) (65.5 mm). The kernel density estimation curve for Glimpse-HR32 is markedly sharper and shifted leftward compared to that of CA-PF-HR32, which exhibits a broader, right-skewed distribution. This quantitative analysis of the error distribution indicates that our method enhances both accuracy and prediction consistency under occlusion. More robustness evaluations with noisy input are provided in the Appendix.

### 4.4. Performance Comparison on MPI-INF-3DHP

Table 2 presents results on the challenging MPI-INF-3DHP dataset. Our Glimpse-HR48 attains a PCK of 98.7%, an AUC of 80.9, and an MPJPE of 29.4 mm, establishing a new state of the art across all metrics among single-frame methods. It achieves a substantial improvement over the leading baseline CA-PF-HR48 (Zhao et al., 2023), exceeding its AUC by 4.6 points (80.9 vs. 76.3) and reducing MPJPE by 2.0 mm (29.4 vs. 31.4). Furthermore, Glimpse-HR48 consistently outperforms the recent PandaPose (Zheng et al., 2026), with superior scores in PCK (+0.1%), AUC (+5.1 points), and MPJPE (-2.4 mm). This strong performance demonstrates the superior generalization capability of our Glimpse to complex real-world scenarios.

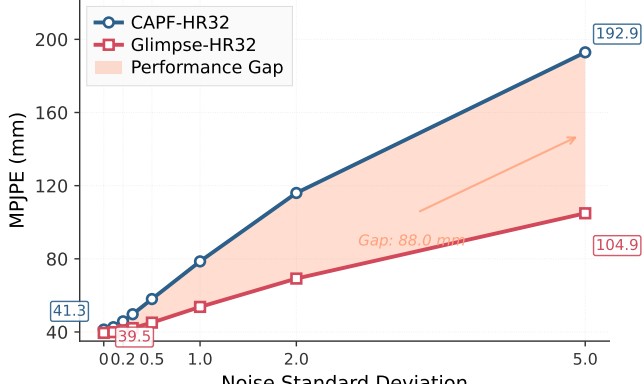

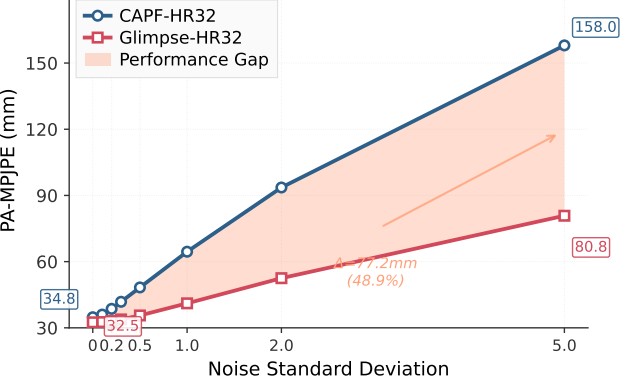

*Figure 4.* Noise robustness comparison on MPJPE and PA-MPJPE metrics on Human3.6M.

### 4.5. Cross-Dataset Evaluation on 3DPW

Table 3 reports cross-dataset results on 3DPW (Von Marcard et al., 2018), a challenging in-the-wild dataset with outdoor scenes and diverse activities. Both methods use 2D poses from HRNet and are trained on Human3.6M. Glimpse-HR32 achieves 132.2 mm MPJPE and 89.0 mm PA-MPJPE, outperforming CA-PF-HR32 by 6.7 mm and 1.1 mm respectively, confirming that structured sampling and geometric correction learn generalizable structural priors rather than overfitting to the source domain.

### 4.6. Robustness Analysis

To evaluate robustness against imprecise 2D keypoint inputs, we simulate Gaussian noise on the input keypoints at eight levels: $\sigma = \{0, 0.1, 0.2, 0.3, 0.5, 1.0, 2.0, 5.0\}$ pixels. The noise is applied independently to each keypoint's $(x, y)$ coordinates, simulating common detection errors in real-world scenarios. We compare our Glimpse-HR32 against the baseline CA-PF-HR32 (Zhao et al., 2023) method using both MPJPE and P-MPJPE as evaluation metrics.

As shown in Figure 4, Glimpse-HR32 demonstrates superior noise robustness compared to CA-PF-HR32 across

*Table 4.* Ablation study on Human3.6M. JS: Joint-level Sampling, LS: Limb-level Sampling, GC: Geometric Correction. $M$ denotes the number of intermediate points sampled per limb. The best result in each sub-table is shown in **bold**.

| (a) Ablation of core components | | | |
|---|---|---|---|
| Method | JS | LS | GC | MPJPE $\downarrow$ |
| Baseline | | | | 51.2 |
| + Joint Sampling | ✓ | | | 41.4 |
| + Joint + Limb Sampling ($M = 3$) | ✓ | ✓ | | 40.3 |
| + Joint Sampling + GC | ✓ | | ✓ | 40.3 |
| + Full Model (JS + LS + GC) | ✓ | ✓ | ✓ | **39.5** |
| (b) Ablation on number of limb points ($M$) | | | |
| + Joint + LS ($M = 1$) | ✓ | ✓ | | 40.6 |
| + Joint + LS ($M = 2$) | ✓ | ✓ | | 40.5 |
| + Joint + LS ($M = 3$) | ✓ | ✓ | | **40.3** |
| + Joint + LS ($M = 4$) | ✓ | ✓ | | 40.4 |

*Table 5.* Computational cost and accuracy ablation.

| JS | LS | GC | MPJPE | Para. (M) | FLOPs (G) | Time (ms) | FPS |
|---|---|---|---|---|---|---|---|
| | | | 51.2 | 13.74 | 7.99 | 30.96 | 32.30 |
| ✓ | | | 41.4 | 14.09 | 8.03 | 33.71 | 29.67 |
| ✓ | ✓ | | 40.3 | 14.12 | 8.06 | 50.18 | 19.93 |
| ✓ | | ✓ | 40.3 | 14.29 | 8.04 | 40.32 | 24.80 |
| ✓ | ✓ | ✓ | **39.5** | 14.32 | 8.09 | 55.87 | 17.90 |

both MPJPE and P-MPJPE metrics. Under zero noise, Glimpse-HR32 achieves 39.5 mm MPJPE and 32.5 mm P-MPJPE, outperforming CA-PF-HR32 (41.3 mm and 34.8 mm). More importantly, under strong noise ($\sigma = 5.0$ pixels), Glimpse-HR32 shows only 104.9 mm MPJPE and 80.8 mm P-MPJPE, significantly outperforming CA-PF-HR32 by 88.0 mm (45.6% improvement) and 77.2 mm (48.9% improvement), respectively. The widening performance gap with increasing noise demonstrates Glimpse's practical advantage for handling noisy keypoint inputs in real-world scenarios.

### 4.7. Ablation Study

In Table 4(a), we conduct ablations on the Glimpse-HR32 variant to validate the contribution of each component. Starting from a baseline that lifts only sparse 2D coordinates to 3D without any visual features (MPJPE = 51.2 mm), adding joint-level deformable sampling reduces error to 41.4 mm, confirming the necessity of extracting adaptive local appearance features. Incorporating limb-level sampling ($M = 3$) further improves performance to 40.3 mm. This gain directly validates our core motivation: explicitly modeling the continuous geometry of limbs provides indispensable structural context that is missing from isolated joint features. Notably, adding geometric correction alone also achieves a comparable reduction (40.3 mm), affirming its role in establishing a canonical 3D reference to ensure global pose consistency. The full model, which integrates both struc-

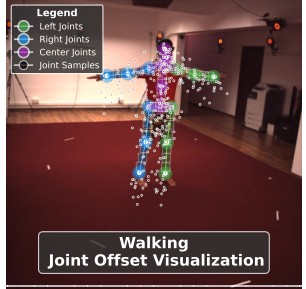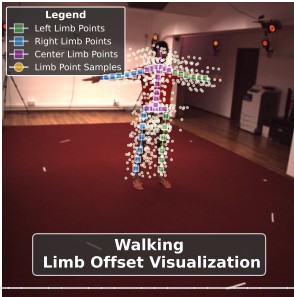
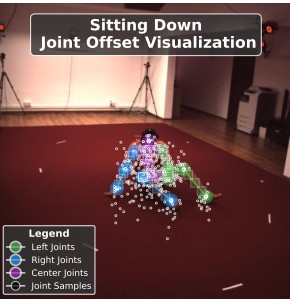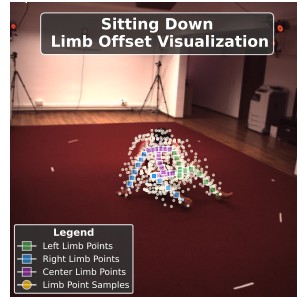

*Figure 5.* Visualization of learned sampling points. Top: "Walking" action. Bottom: "Sitting Down" action. Left: joint-level sampling. Right: limb-level sampling.

tured sampling for 2D continuity and geometric correction for 3D consistency, achieves the best result of 39.5 mm. These systematic improvements validate the efficacy of each proposed component in our Glimpse.

In Table 4(b), we ablate the number of intermediate limb points $M$. Performance peaks at $M = 3$ (40.3 mm), with degradation at $M = 4$, indicating that three points optimally balance geometric coverage against over-sampling. This result reflects a trade-off: too few points ($M < 3$) fail to adequately capture the limb's continuous structure, while too many ($M > 3$) introduce redundant or noisy features. This validates our design choice of $M = 3$.

In Table 5, we quantify the computational cost of each proposed component. The baseline model achieves 51.2 mm MPJPE at 32.30 FPS. Joint Sampling reduces MPJPE by 9.8 mm with only 1.3 ms overhead. Adding Limb Sampling with M=3 further improves accuracy while introducing moderate latency. Geometric Correction incurs 15.5 ms but contributes 0.8 mm improvement by aligning multi-scale features with the canonical reference. The full Glimpse-HR32 achieves 39.5 mm MPJPE at 17.90 FPS, maintaining a practical frame rate for real-time applications while substantially outperforming the baseline.

### 4.8. Qualitative Analysis

**Visualization of Structured Sampling** Figure 5 visualizes the learned sampling locations on Human3.6M. The results show anatomically coherent patterns: joint-level off-

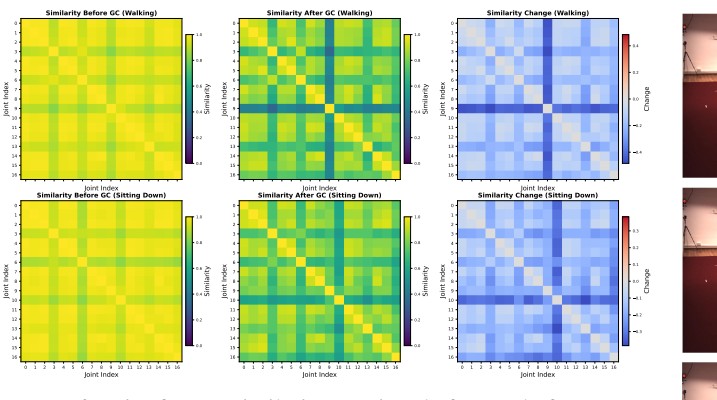

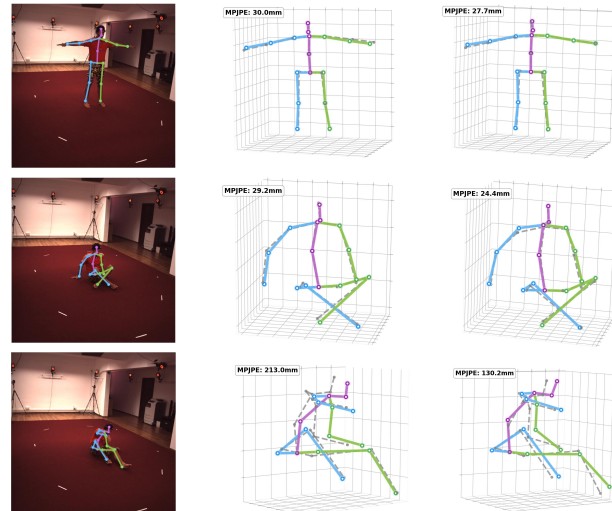

*Figure 6.* Joint feature similarity matrices before and after geometric correction for Walking (top) and Sitting Down (bottom) actions. Indices of rows and columns correspond to the 17 joints: Pelvis, Right Hip, Right Knee, Right Ankle, Left Hip, Left Knee, Left Ankle, Spine, Thorax, Neck, Head, Left Shoulder, Left Elbow, Left Wrist, Right Shoulder, Right Elbow, Right Wrist.

*Figure 7.* Qualitative comparison of 3D-HPE results on three poses. From left to right: input 2D pose (projected onto the image), output of the CA-PF-HR32 method (Zhao et al., 2023), and output of our Glimpse-HR32 method, with corresponding MPJPE values annotated. Ground-truth 3D poses are shown in gray for reference.

sets (left column) form tight clusters around estimated key-points, focusing on local appearance. In contrast, limb-level sampling (right column) distributes points along the interpolated skeletal segments and, crucially, also spreads laterally across the limb width, capturing mid-limb appearance and directional context. This dual behavior is consistent across "Walking" (top row) and the heavily occluded "Sitting Down" (bottom row), where sampling points maintain structural coherence even when limbs overlap or are partially hidden. The visualization thus illustrates how structured sampling overcomes the limitation of point-wise features, as it explicitly respects the body's topological constraints while adaptively gathering structural visual evidence.

**Impact of Geometric Correction** Figure 6 visualizes the pairwise cosine similarity between joint features before and after geometric correction. The initial features, derived primarily from the 2D pose embeddings, exhibit high similarity between anatomically distinct joints that are proximate in the image plane, such as the left hip and left wrist. This pattern reflects the inherent ambiguity in monocular lifting, where different 3D configurations can project to similar 2D locations. After geometric correction, the updated pose embeddings (obtained by fusing the geometrically corrected multi-scale image evidence via Eq. 19) yield more structured similarity matrices. Off-diagonal similarities between unrelated joints are reduced, while within-limb relationships remain preserved. This transformation demonstrates that the geometric correction module successfully integrates the canonical 3D structural prior (via the anchor $\mathbf{r}$) into the pose representation, thereby enhancing its discriminability. This, in turn, facilitates more accurate 3D pose regression by distinguishing joints based on their anatomical roles rather than 2D spatial proximity.

**Pose Estimation Visualization.** Figure 7 shows qualita-

tive comparisons on three representative poses from Human3.6M. Glimpse produces more anatomically plausible predictions than CA-PF (Zhao et al., 2023), particularly under self-occlusion, where visible limb regions constrain occluded joints through the skeletal graph. In the most challenging cases involving complex articulation (bottom two rows), CA-PF produces distorted limbs and joint misplacements, whereas Glimpse maintains correct geometric relationships and structural coherence. These visual improvements are reflected in substantially lower MPJPE (e.g., 136.2 mm vs. 213.0 mm). The results highlight the advantage of structured sampling in leveraging continuous limb context to handle occlusion, while geometric correction maintains global consistency via a canonical 3D reference.

## 5. Conclusion

This paper addresses the challenge of occlusion fragility in single-frame 3D-HPE by introducing explicit geometry learning from single images. We propose Glimpse, a framework that learns coherent structural priors through two core innovations: (1) structured sampling, which captures continuous limb geometry via dense feature sampling along skeletal segments, overcoming the prevalent point-wise paradigm; and (2) geometric correction, which projects these features into a canonical 3D space to enforce global consistency. Extensive evaluations on Human3.6M and MPI-INF-3DHP demonstrate that Glimpse achieves state-of-the-art performance and exhibits strong robustness against both occlusion and input noise, thereby validating the efficacy of explicit geometry learning from images.

## Acknowledgments

This work was supported in part by the Science and Technology Development Fund, Macau SAR, under Grant 0193/2023/RIA3 and 0079/2025/AFJ, and the University of Macau under Grant MYRG-GRG2024-00065-FST-UMDF.

## Impact Statement

This paper presents work whose goal is to advance the field of Machine Learning. There are many potential societal consequences of our work, none which we feel must be specifically highlighted here.

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

# A. Appendix

This appendix provides supplementary materials to further validate the effectiveness, robustness, and efficiency of the proposed Glimpse framework. We present: (1) detailed analysis of the learnable gating mechanisms; (2) additional visualizations; and (3) implementation details. All experiments follow the same protocol as described in the main paper.

## A.1. Analysis of Channel-wise Gating Mechanisms

We analyze the behavior of the two channel-wise learnable gating parameters in our framework: the joint-limb fusion gate $\alpha_c$ (Eq. (14)) and the geometric correction gate $\beta_c$ (Eq. (18)), where $c \in \{1, \ldots, d\}$ indexes the feature dimension $d = 128$.

**Gating Parameter Analysis**   Figure 8 shows distributions of channel-wise gating parameters $\alpha_c$ and $\beta_c$ after convergence. Channels exhibit diverse specialization: **2.3%** have $\alpha_c > 0.6$ (joint-emphasizing), **10.2%** have $\alpha_c < 0.48$ (limb-emphasizing), **20.3%** have $\beta_c > 0.55$ (pose-preserving), and **24.2%** have $\beta_c < 0.45$ (correction-emphasizing). Notably, 75% of $\alpha_c$ values lie within 0.48-0.55, indicating balanced joint-limb fusion, while $\beta_c$ shows broader specialization. This heterogeneity enables adaptive information routing across channels.

**Impact of Gating Specialization**   To verify the contribution of learned gating patterns, we conducted an ablation study where we replaced the learned $\alpha_c$ and $\beta_c$ with uniform values ($\alpha_c = \beta_c = 0.5, \forall c$) during inference. This resulted in a performance degradation of 0.5 mm MPJPE on Human3.6M, confirming that the channel-wise specialization learned by the model contributes meaningfully to pose estimation accuracy.

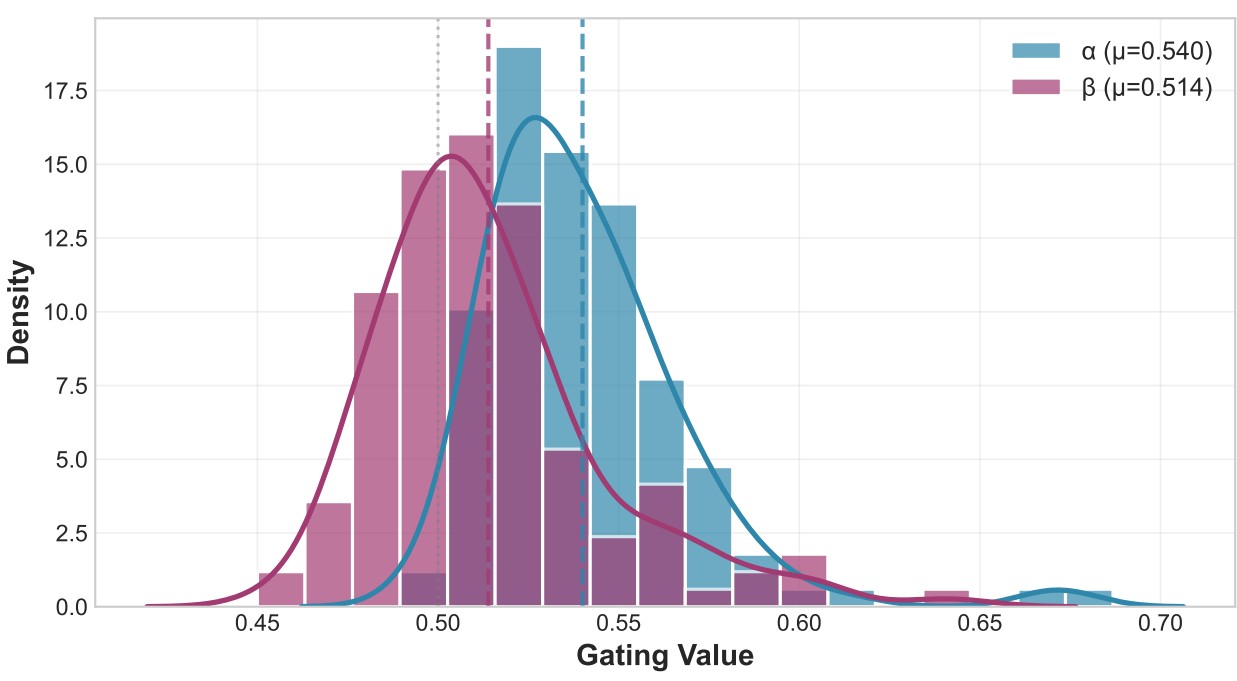

*Figure 8.* Distribution of converged gating parameters $\alpha$ (joint-limb fusion) and $\beta$ (geometric correction) across all feature channels. Both distributions are shown as overlaid histograms with kernel density estimates. The $\alpha$ distribution (blue) centers at 0.540, and $\beta$ (red) centers at 0.514.

## A.2. Additional Qualitative Results

### A.2.1. Structured Sampling Visualizations

We provide additional visualizations of the learned sampling points for the *SittingDown* action from subject S9 in Human3.6M (frames 400, 600, 800, 1000, 1200). As shown in Figure 9, the sampling patterns adapt to the underlying skeletal topology:

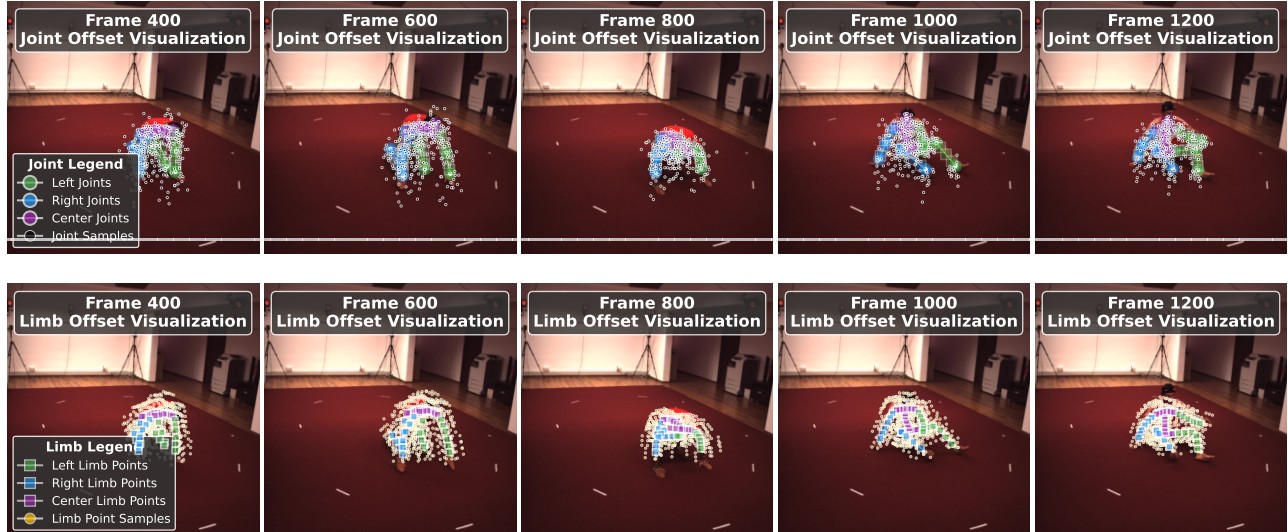

*Figure 9.* Additional visualizations of structured sampling for *Sitting Down* actions. Top: joint-level sampling. Bottom: limb-level sampling.

joint-level sampling remains localized around keypoints, while limb-level sampling distributes along and across skeletal segments.

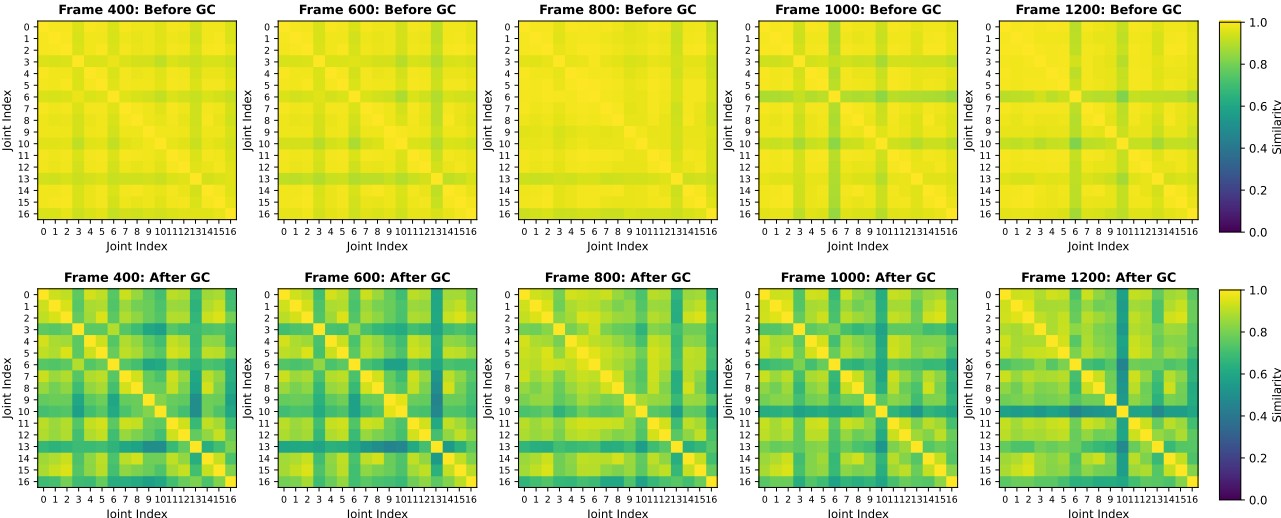

*Figure 10.* Joint feature similarity matrices before (top) and after (bottom) geometric correction in a SittingDown sequence. Rows represent different frames from the motion.

### A.2.2. FEATURE SIMILARITY ANALYSIS

We analyze the feature similarity in the *SittingDown* action performed by subject S9. As shown in Figure 10, the geometric correction module adjusts joint feature relationships throughout the sitting motion. The similarity matrices show reduced off-diagonal values (indicating suppressed spurious correlations) and enhanced diagonal coherence after correction across different frames (400, 600, 800, 1000, 1200) of the sequence.

### A.2.3. 3D HUMAN POSE ESTIMATION RESULTS

Figure 11 presents qualitative comparisons on Human3.6M for subjects S9 and S11 performing various actions. Our Glimpse consistently produces more anatomically plausible 3D poses compared to CA-PF across different motion types.

Figure 12 presents additional qualitative results on the MPI-INF-3DHP test set with challenging outdoor scenes and severe occlusions. Our Glimpse method demonstrates robustness in estimating accurate 3D poses under self-occlusion and extreme viewpoints.

## A.3. Implementation Details

### A.3.1. MODEL ARCHITECTURE

The complete architecture of Glimpse is implemented in PyTorch 1.12.0. The model is trained on a single NVIDIA A6000 GPU. We train Glimpse for 60 epochs using the Adam optimizer with an initial learning rate of 0.001, which decays by a factor of 0.1 at epochs 30 and 50. The batch size is set to 512.

The structured sampling module uses $K = 4$ sampling points per joint location and $M = 3$ intermediate points per limb. The geometric correction module employs a temperature parameter $\sigma$ set to 1.0. The feature dimension $d$ is set to 128 across all components.

### A.3.2. CODE AVAILABILITY

The PyTorch implementation of the Glimpse framework is included in the supplementary material. This submission contains the core model implementation. The full source code and pre-trained models will be made publicly available upon acceptance to ensure reproducibility and facilitate further research.

CAPF    Glimpse    CAPF    Glimpse

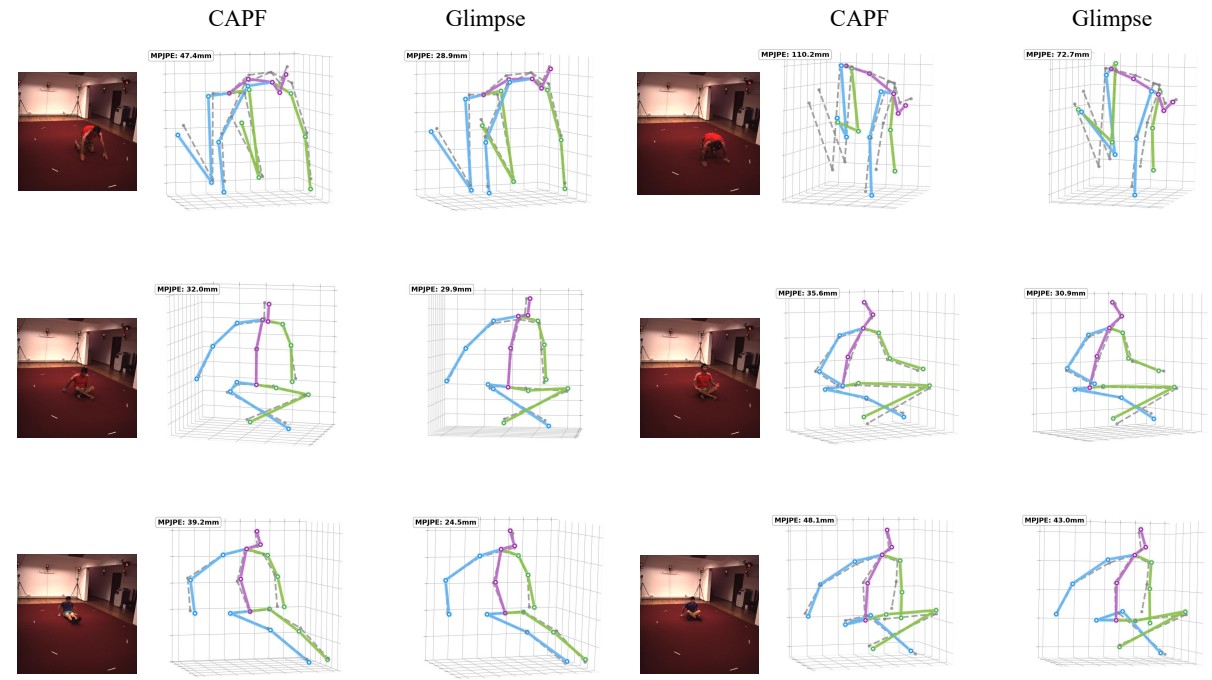

*Figure 11.* Qualitative comparisons on Human3.6M with subjects S9 and S11. Columns from left to right: input image with 2D pose overlay, CA-PF prediction, our Glimpse prediction, and ground truth (gray skeleton).

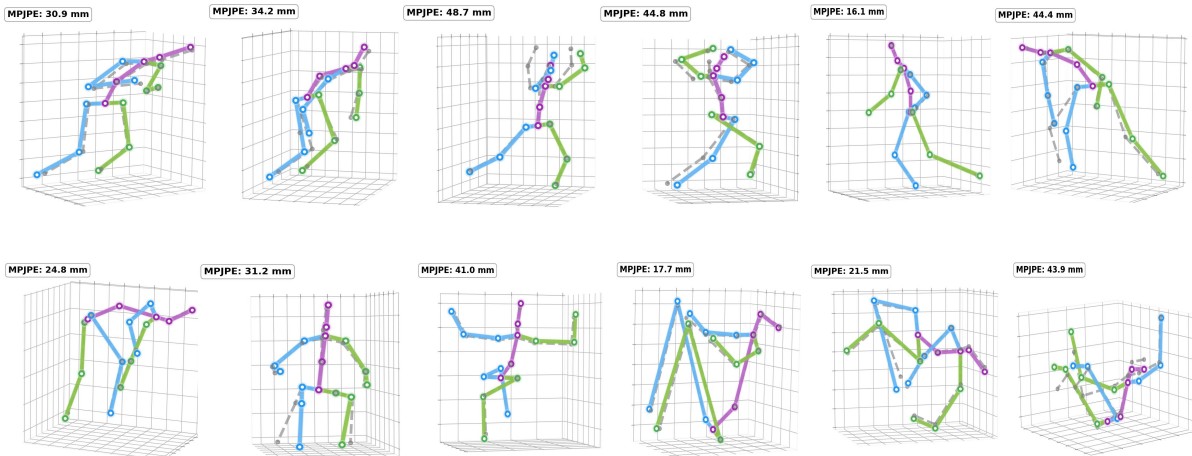

*Figure 12.* Qualitative results on the MPI-INF-3DHP test set. Each subfigure presents a distinct challenging sample (TS1TS6), comparing our Glimpse prediction (colored skeleton) against the ground truth (gray skeleton).

