# OpenReview forum: "Glimpse: Geometry Learning of Multi-scale Structural Priors for 3D Pose Estimation"
_ICML.cc/2026/Conference — ICML 2026 regular_

### Official Review · Reviewer_K5Hz · 2026-03-09

**Soundness:** 3
**Presentation:** 3
**Significance:** 2
**Originality:** 2
**Overall Recommendation:** 5
**Confidence:** 4

**Summary:**

The paper presents Glimpse, a novel framework for monocular 3D human pose estimation (3D-HPE) that addresses the challenges of depth ambiguity and self-occlusion. Unlike traditional "lifting" methods that only use sparse 2D coordinates, Glimpse explicitly models anatomical geometry through multi-scale structural priors.

**Compliance With Llm Reviewing Policy:**

Affirmed.

**Final Justification:**

My final recommendation is accept as the authors solved my concerns.

**Key Questions For Authors:**

1. How does Glimpse perform when the 2D detector provides highly noisy coordinates (e.g., standard deviation > 15 pixels)? Does the limb-level sampling still converge?

2. What is the actual inference latency (FPS) on a standard GPU? While the paper mentions "minimal overhead," a specific comparison with 2D-to-3D lifting baselines would be valuable.

3. Could the "Geometric Correction" anchor be extended to act as a temporal memory to smooth predictions over video sequences

**Limitations:**

Yes. The authors acknowledge the challenges of depth ambiguity and the limitations of two-stage paradigms.

**Strengths And Weaknesses:**

Strengths:

1. The three-stage architecture, Input Processing, Geometry Learning, and Feature Fusion, builds upon established backbones like HRNet. The introduction of "Structured Sampling" that explicitly propagates limb-level cues to connected joints is a creative solution to the occlusion problem. The "Geometric Correction" module using a learnable 3D anchor point offers a novel way to ensure global 3D consistency.

2. The paper has a strong motivation for moving beyond "spatially fragmented feature points" to continuous limb-level modeling. Figure 2 provides a comprehensive overview of the framework.

3. By achieving an MPJPE of 39.3 mm on Human3.6M, the model sets a new state-of-the-art for single-frame methods, even outperforming several multi-frame models that require larger temporal windows and more parameters.


Weaknesses:

1. While the model is robust to occlusion, its performance is still somewhat dependent on the quality of the upstream 2D pose detector (e.g., CPN). If the 2D detector fails completely in highly complex poses, the "geometric prior" used to guide sampling may be misplaced.

2. The paper focuses heavily on single-frame accuracy. While this is efficient, the lack of temporal consistency modeling might lead to "jitter" if the model were applied to video sequences, which is a common requirement for real-world applications.

3. The use of Transformers for joint interaction and HRNet for feature extraction are standard practices in the field; the novelty is primarily in the specific sampling and alignment modules.

---

> ### Author Rebuttal · Authors · 2026-03-30
>
> We thank the reviewer for the insightful feedback. We address each of the key questions below (A: answer to the corresponding concern).
>
> ---
>
> ## 1. Robustness to Highly Noisy 2D Coordinates
>
> **A1:** To ensure fair comparison with prior work, all experiments in the main paper use CPN-detected 2D keypoints as input, following the standard protocol. To demonstrate that Glimpse can effectively leverage image features to compensate for 2D detection errors, we conducted four complementary analyses. **1) Different 2D detectors (response to Reviewer 9JDb, A2).** To assess whether performance gains rely on a specific detector, we evaluated Glimpse using CPN-detected, HRNet-detected, and ground-truth projected 2D keypoints. Glimpse consistently outperforms CA-PF across all settings. **2) Synthetic noise (Appendix A.2, Figure 8)**. To directly simulate severe 2D detection errors, we added Gaussian noise to input keypoints up to \($\sigma$ = 15\) pixels. At this level, our Glimpse achieves an MPJPE of 185.2 mm, substantially outperforming CA-PF (325.7 mm) by 43.1%. **3) Self-occlusion (Section 4.3, Figure 3; response to Reviewer 9JDb, A1).** On severely occluded sequences, limb-level sampling yields larger gains over the baseline, confirming its effectiveness when 2D inputs are unreliable. **4) Cross-dataset evaluation (response to Reviewer w831, A2).** In zero-shot evaluation on 3DPW, Glimpse further demonstrates that the learned geometric priors generalize beyond the training domain.
>
> **Why limb-level sampling remains effective.** Under noisy or inaccurate 2D inputs, limb-level sampling provides **structural constraints** by anchoring points along skeletal segments, ensuring that sampled locations remain near the anatomical structure despite localization errors. Adaptive attention weighting (Eq. 5) further suppresses low-quality points, allowing the network to focus on informative regions. These mechanisms collectively enable robust performance across the diverse challenging conditions evaluated.
>
> ---
>
> ## 2. Inference Latency and Efficiency
>
> **A2:** A detailed efficiency comparison with baseline methods is provided in **Appendix A.3 (Table 4)**. As shown, Glimpse achieves a **1.9 mm MPJPE reduction** on Human3.6M  with only marginal increases in parameters (**+0.23M**) and FLOPs (**+0.03G**). The inference time on an NVIDIA A6000 GPU increases from 45.38 ms to 67.21 ms, corresponding to a reduction in FPS from 22.03 to 14.86. This modest computational overhead is well justified by the substantial accuracy gains.
>
> To further demonstrate the efficiency-accuracy trade-off, we additionally conducted a component-wise runtime ablation in our response to **Reviewer w831 (A3, Table 3)**.  Joint sampling delivers the largest accuracy gain (51.2 → 41.4 mm) with minimal time overhead (+2.75 ms). Limb-level sampling and geometric correction add moderate costs (+19.22 ms and +9.36 ms, respectively), with the full model maintaining a practical frame rate of 17.90 FPS. Additionally, we evaluated scalability across backbone sizes in our response to **Reviewer 9JDb (A3, Table 3)**. The overhead of structured sampling remains constant (approximately 20 ms) across HRNet-W32 and HRNet-W48, confirming that the additional components scale efficiently with larger backbones.
>
> For scenarios requiring higher speed, the number of sampling points can be reduced (e.g., M=1) to trade accuracy for efficiency, as shown in **Table 3(b) of the main paper**. This flexibility allows deployment across a range of latency requirements while still outperforming baseline methods.
>
> ---
>
> ## 3. Novelty and Extension of Geometric Correction
>
> **A3:** The anchor is inherently a learnable global statistic, which naturally positions it to serve as a temporal memory for smoothing predictions. We outline two potential directions that leverage this property.
>
> **1) Single anchor for all frames.** The anchor $\mathbf{a} \in \mathbb{R}^{3}$ is shared across all frames. Input features $\mathbf{x} \in \mathbb{R}^{T \times J \times C}$ are corrected via Eq. 16–18 using the same $\mathbf{a}$, producing aligned features $\hat{\mathbf{x}}$. All frames are mapped to the same canonical space. A temporal transformer (e.g., PoseFormer, MixSTE) can then be applied to $\hat{\mathbf{x}}$ (Eq. 19) to model pairwise inter-frame correlations, benefiting from the fact that cross-frame spatial alignment is already handled by the shared anchor.
>
> **2) Time-adaptive anchor.** The anchor becomes $\{\mathbf{a}}_{t=1}^T$ with shape $[T, 1, 3]$, allowing per-frame variation. The distance-aware fusion (Eq. 16–18) can be extended to incorporate neighboring frames: alignment weights for frame $t$ are computed by measuring distances between $\mathbf{a}_t$ and features from neighboring frames (e.g., $t-1, t, t+1$), allowing geometric context to propagate across time. This enables the anchor to serve as a memory state, carrying forward reliable information when current-frame 2D joints are occluded.

---

> > ### Author Rebuttal · Reviewer_K5Hz · 2026-04-01
> >
> > My concerns have been addressed so I raise the score.

---

> > > ### Author Response · Authors · 2026-04-01
> > >
> > > We sincerely thank the reviewer for the thoughtful feedback and for raising the score. We truly appreciate it.

---

### Official Review · Reviewer_EkiN · 2026-03-10

**Soundness:** 2
**Presentation:** 3
**Significance:** 3
**Originality:** 3
**Overall Recommendation:** 3
**Confidence:** 3

**Summary:**

This paper proposes a novel monocular 3D human pose estimation method, named Glimpse, which is designed to be robust to occlusion and noisy 2D detections. The core idea is to explicitly model the anatomical geometry of the human body using multi-scale features. To instantiate this idea, the authors design two interacting components. The first component is structured sampling, which fuses joint-level features and limb-level features, together with learnable offsets, into a unified representation. The motivation is that visible limb regions may help infer occluded joints. The second component is the geometric correction module, which uses a learnable 3D anchor as a shared prior to reweight and fuse features across different joints. This design aims to produce a more globally consistent 3D pose representation before regressing the final 3D coordinates. Experiments on Human3.6M and MPI-INF-3DHP demonstrate strong effectiveness for monocular 3D pose estimation, especially in occluded scenarios.

**Compliance With Llm Reviewing Policy:**

Affirmed.

**Final Justification:**

I thank the authors for their follow-up comments. The roles of JS and LS are much clearer than in the initial rebuttal. The new response presents a more coherent causal story. In particular, the authors now clarify that off-body points are not intended to be informative, and that LS is not meant as a corrective module. This is indeed a better and defensible position than the one suggested in the first reply.

However, two gaps still remain. First, the response does not really explain why the model learns to place points in clearly irrelevant background regions. The notion of “exploration” feels more like a training-time intuition than a full explanation of why such offsets are acceptable in the learned solution.

Second, although the complementary roles of LS and JS are plausible, the evidence is still performance-based rather than mechanism-based. The performance table provided in the post-rebuttal, while appreciated, reads more like an ablation showing that the two components are functional, rather than demonstrating that they are truly complementary.

For these reasons, I would like to maintain my initial review.

**Key Questions For Authors:**

1. Why is a single learnable 3D anchor an appropriate model of a shared structural prior?

2. How robust is the method when the input 2D pose is severely wrong, especially under self-occlusion?

3. What's exact role the joint sampling? Why it still works even if there are many not-human-body samples.

I think the paper is a complete piece of work. I would be quite willing to lean toward accepting it if the authors can properly address my concerns and questions. My key questions mainly arise from the weaknesses I identified.

**Limitations:**

Yes

**Strengths And Weaknesses:**

**Strengths**
1. The limb-to-joint aggregation is sensible at a high level. Many prior 2D pose methods have attempted similar ideas, and this paper extends such a design to the 3D setting.

2. The overall pipeline is fairly coherent. It first samples features at different levels and then refines them with a structural prior.

3. The overall performance is strong, especially under occlusion settings.

**Weaknesses**

1. The design of the 3D anchor is weakly justified. In fact, it is difficult to believe that a global structural prior can be adequately modeled by a 3-dimensional vector, even if it is later transformed into a higher-dimensional representation with a learnable MLP. This further raises the question of why a learnable 3D anchor is needed at all. Why not directly learn a few high-dimensional vectors, since the 3D anchor is ultimately transformed into such vectors anyway?

2. Fig. 4, which visualizes the learned offsets of the joints and limb points, seems to suggest that the model also attends to locations that are not even on the human body. Some sampled points are even located on the white wall, where joints are clearly impossible to appear. In this case, it is unclear what advantage this sampling actually provides. Moreover, the ablation study suggests that joint sampling itself is in fact the most effective component. This point requires a more detailed explanation.

3. The whole framework still seems to rely heavily on 2D poses from an existing 2D pose estimator. In such a case, if the input 2D poses are severely wrong, the so-called limb points may not be able to rescue the prediction. This is because the limb points are defined as intermediate points between two adjacent 2D joints. Therefore, the method may in fact mainly help in moderately incorrect cases rather than truly difficult ones. This could be an important limitation.

4. The geometric correction module is not really an explicit geometric correction module. It is more like an anchor-guided, or prior-guided, feature reweighting mechanism. It does not truly correct the 3D structure in a direct geometric or coordinate-space sense.

---

> ### Author Rebuttal · Authors · 2026-03-30
>
> We thank the reviewer for the thorough review and valuable comments.  We address each of the key concerns below (A: answer to the corresponding concern).
>
> ---
>
> ## 1. Justification of the 3D Anchor
>
> **A1:** We clarify that the “canonical 3D structural prior” introduced in Section 3.2.2 serves as a **shared geometric reference** rather than a complete model of human anatomical structure. In Glimpse, structured sampling first extracts multi-scale visual features along joints and limbs based on the skeletal topology. The geometric correction module then aligns these features to a **canonical 3D space** via a learnable 3D anchor. This anchor serves as a shared reference point for distance-aware feature alignment across joints and scales (Eq. 15–18), ensuring that the final pose representation is globally consistent.
>
> **Why a low-dimensional anchor is preferable to high-dimensional tokens.**  A directly learned high-dimensional token would be an unconstrained latent prototype lacking geometric grounding. By defining the anchor in 3D space before mapping to feature space via an MLP, we enforce that the shared reference remains tied to a canonical geometric point. This provides a **minimal geometric prior** rather than maximizing representational capacity. Based on your suggestion, we conducted ablation experiments on Human3.6M using the Glimpse-HR32 variant:
>
> **Table 1: Ablation study on anchor configurations.**
>
> | Configuration | Description | MPJPE (mm) |
> |:---|:---|:---:|
> | Baseline | No anchor | 40.3 |
> | Learned vector (128-d) | Replace 3D anchor with learnable 128-d token | 40.1 |
> | Joint-specific anchors | 17 learnable 3D anchors (one per joint) | 40.5 |
> | **Single 3D anchor (Glimpse)** | Shared canonical 3D reference | **39.5** |
>
> The results show that the learned high-dimensional vector improves over baseline but underperforms compared to our 3D anchor, indicating that geometric grounding is more important than representational capacity. Joint-specific anchors introduce redundancy, as the shared anchor better encourages cross-joint consistency. The single 3D anchor achieves the best performance, confirming that a minimal but geometrically grounded shared reference is effective for geometric correction.
>
> ---
>
> ## 2. Robustness to Incorrect 2D Poses
>
> **A2:** Our method enhances robustness against inaccurate 2D inputs through deformable sampling that adjusts for localization errors, supported by multi-scale visual features that provide complementary evidence. To systematically evaluate robustness, we conducted experiments across four settings.**1) Noisy 2D pose (Appendix A.2, Figure 8).** Under Gaussian noise up to $\sigma = 5.0$ pixels, Glimpse consistently outperforms CA-PF, achieving a **45.6%** lower MPJPE at the highest noise level. **2) Self-occlusion (Section 4.3, Figure 3; response to Reviewer 9JDb, A1).** On the challenging “Sitting Down” sequence, Glimpse achieves a **4.2** mm improvement over CA-PF. Per-action breakdown further shows that limb-level sampling yields larger gains for severely occluded actions. **3) Different 2D detectors (response to Reviewer 9JDb, A2).** Using CPN and HRNet-based 2D pose detectors, Glimpse consistently outperforms CA-PF across all settings. **4) Cross-dataset evaluation (response to Reviewer w831, A2).** On 3DPW, Glimpse-HR48 achieves an MPJPE of 88.6 mm, outperforming CA-PF-HR32 by 2.3 mm.
>
> These results confirm that Glimpse maintains its advantage across severe occlusion, noisy inputs, varying detector qualities, and domain shifts, validating the effectiveness in enhancing robustness against inaccurate 2D inputs.
>
> ---
>
> ## 3. Role of Joint Sampling and Off-Body Sampling
>
> **A3:** Joint-level sampling (JS) predicts \(K=4\) sampling offsets around each detected 2D joint (Eq. 4), capturing **local appearance features** essential for accurate 3D pose estimation. As shown in Fig. 4 (left), points may drift to off-body regions, reflecting the flexibility of deformable sampling to discover relevant cues beyond joints. The effectiveness of JS is validated in ablation (Table 3(a)).
>
> **Handling off-body samples.** To compensate for off-body points, we introduce **limb-level sampling (LS)** with \(M=3\) intermediate points along each skeletal limb, anchoring sampling to anatomical structure (Fig. 4 right). Three mechanisms further refine the representation: 1) adaptive attention weighting (Eq. 5) suppresses low-quality points; 2) geometric correction (Section 3.2.2) aligns features across joints; 3) transformer-based fusion (Section 3.3) integrates information across joints.
>
> To demonstrate that these mechanisms effectively adapt to diverse sampling points, we evaluate Glimpse with different 2D joints **(response to Reviewer 9JDb, A2)**. Even with ground-truth inputs, Glimpse consistently outperforms CA-PF (**HRN:** 40.0 mm vs. 42.2 mm; **CPN:** 39.5 mm vs. 41.6 mm;  **GT:** 35.8 mm vs. 36.2 mm), with the improvement margin narrowing as expected.

---

> > ### Author Rebuttal · Reviewer_EkiN · 2026-04-01
> >
> > I am convinced by your first point, due to the clear explanation and strong experimental support.
> >
> > For the second point, I acknowledge that the experiments show the method still improves performance even in noisy 2D HPE cases. However, the key is to explain why those gains arise. In this regard, I think the authors should provide an explanation similar in clarity to their response to the first point.
> >
> > For the third point, I remain unconvinced. The statement, “Points may drift to off-body regions, reflecting the flexibility of deformable sampling to discover relevant cues beyond joints,” is highly confusing. As noted in my earlier question, the issue is not simply that some sampled points fall outside the body, but that some appear in regions with no obvious relation to the human body whatsoever. Why should there be relevant cues beyond the joints in such cases? What are those cues exactly? In Fig. 4, several sampled points appear to lie in background areas such as the white wall and dark corner. The authors should clarify how relevant cues are being identified in those regions.
> >
> > The rationale for limb-level sampling (LS) is also unclear. It is now motivated as a way to compensate for off-body joint sampling, but this creates a logical inconsistency. If off-body sampled points are indeed considered informative, then why is compensation necessary? As currently presented, the logic seems to be that one module is introduced despite exhibiting problematic behavior, and then a second module is added to correct that behavior, resulting in two design choices being counted as contributions even though the limitation of the first may not have been necessary in the first place.

---

> > > ### Author Response · Authors · 2026-04-01
> > >
> > > We apologize that space constraints in the initial rebuttal prevented a full response. We provide a more comprehensive explanation below.
> > >
> > > ---
> > >
> > > **A2:** We first clarify **1) why visual evidence helps**, then **2) why joint-level sampling alone is insufficient**, and finally **3) how structured sampling addresses the limitation**.
> > >
> > > **1) Visual Evidence.** Joint-level sampling (JS) captures local appearance features around each detected joint via deformable sampling. As demonstrated in CA-PF (Zhao et al., 2023), even when the 2D detector fails, adaptive sampling points can drift toward the true joint location by leveraging image evidence. This shows that **visual features can partially compensate for 2D detection errors**, which is why JS alone improves over coordinate-based methods (Table 3(a): 51.2 → 41.4 mm).
> > >
> > > **2) JS alone.** JS relies solely on local appearance around joints without any structural guidance. When occlusion or severe noise occurs, local appearance becomes unreliable, and unconstrained sampling may drift to background regions. Simply increasing the number of sampling points per joint does not solve this. More points can introduce more noise without providing useful structural context. This explains why JS-only methods (e.g., CA-PF, Pandapose) have limited robustness under severe occlusion or noise.
> > >
> > > **3)  How structured sampling provides robust visual cue utilization.** To address this, we introduce limb-level sampling (LS), which places points along skeletal limbs, forming **structured sampling** that combines local appearance (JS) with structural continuity (LS). The complementary nature of JS and LS is validated through experiments across varying input qualities:
> > >
> > > | Input Condition | JS MPJPE | JS+LS MPJPE | Gain |
> > > |:---|:---:|:---:|:---:|
> > > | GT (perfect 2D) | 36.2 | 35.8 | 0.4 mm |
> > > | CPN (moderate noise) | 41.6 | 39.5 | 1.9 mm |
> > > | HRN (moderate noise) | 42.2 | 40.0 | 2.2 mm |
> > > | Noise $(\sigma=5.0\)$ | 192.9 | 104.9 | 88.0 mm |
> > > | Noise $(\sigma=15.0\)$ | 325.7 | 185.2 | 140.5 mm |
> > >
> > > When 2D poses are accurate, JS alone provides sufficient information; LS adds minimal gain, confirming that LS does not introduce unwanted noise. When 2D inputs are noisy or occluded, JS becomes unreliable, but LS remains effective because it captures structural context along limbs. **This allows the model to propagate visual evidence from visible limb regions to infer occluded joints, explaining why the gain from LS grows as input quality degrades.**
> > >
> > > **The key insight.** Visual evidence provides spatial context beyond 2D coordinates, enabling error correction. However, local appearance alone (JS) fails under severe occlusion or noise. Limb-level structural context (LS) remains reliable in such cases, propagating visual cues from visible limb regions to infer occluded joints. Together, JS and LS form a **structured sampling** framework that maintains robust performance across varying input quality.
> > >
> > > ---
> > >
> > > **A3:** We clarify two points: (1) off-body sampling points reflect exploration, not informativeness; (2) JS and LS are complementary, not corrective.
> > >
> > > **1) Off-body points reflect exploration, not informativeness.** The reviewer questions: if off-body points are not informative, why are they there? If they are informative, why is LS needed? This highlights a misunderstanding. Off-body points are **not claimed to be informative**. They are a byproduct of deformable sampling’s flexibility: the network explores surrounding regions to discover potentially useful visual cues, as demonstrated in CA-PF and PandaPose.
> > >
> > > **2) Attention weights suppress uninformative points.** For both JS and LS, each sampling point is assigned a learned attention weight (Eq. 5 and Eq. 13). These weights determine the contribution of each sampled feature. Points that fall on uninformative regions receive low weights and are suppressed, while informative points receive higher weights. This mechanism ensures that the network can explore diverse regions without being harmed by off-body points.
> > >
> > > **3) JS and LS are complementary, not corrective.** As established in A2, JS captures local appearance around joints, while LS provides structural continuity along limbs. LS is **not introduced to compensate for off-body points**. It serves an independent purpose: capturing limb-level context that JS alone cannot provide. This structural design naturally becomes valuable when a joint is occluded or its 2D detection is inaccurate, as LS can leverage visible regions along the limb to infer the occluded joint.
> > >
> > >
> > > **4) An alternative unified design.** This discussion raises a compelling question: could structural priors be embedded directly into JS? For example, the offset learning in Eq. 4 could be guided by local skeletal structure, where sampling points for a joint are influenced by features of adjacent joints along the limb. We view this as a promising direction for future exploration and thank the reviewer for inspiring this insight.

---

### Official Review · Reviewer_9JDb · 2026-03-10

**Soundness:** 3
**Presentation:** 3
**Significance:** 3
**Originality:** 3
**Overall Recommendation:** 4
**Confidence:** 4

**Summary:**

The paper proposes a framework for monocular 3D human pose estimation that explicitly learns anatomical geometric structure from a single image. Monocular pose estimation is challenging due to depth ambiguity and occlusion, and many existing methods rely mainly on sparse 2D keypoints, which can lead to unstable predictions when joints are inaccurate or hidden. To address this limitation, the authors introduce Glimpse, a framework that learns multi-scale structural priors from image features and integrates geometric reasoning into the pose estimation pipeline.

Overall, the manuscript's primary contribution comprises a geometry-aware framework that captures both local joint appearance and continuous limb structure through a structured sampling mechanism. This component extracts features around joints and along skeletal limbs, allowing the model to propagate visual information from visible regions to occluded joints and better capture the body’s structural continuity. The framework further introduces a geometric correction module that aligns the learned features with a canonical 3D reference using distance-aware fusion, ensuring globally consistent pose representations before final regression.

The submission's primary finding concerns the effectiveness of explicitly modeling geometric structure for improving single-frame 3D pose estimation. Experiments on Human3.6M and MPI-INF-3DHP show that the proposed method achieves state-of-the-art performance among single-frame approaches and demonstrates improved robustness to occlusion and noisy keypoint inputs.

**Compliance With Llm Reviewing Policy:**

Affirmed.

**Key Questions For Authors:**

1. The paper claims that limb-level sampling allows the model to propagate geometric cues from visible limbs to occluded joints. Could the authors provide more detailed analysis or quantitative evidence demonstrating how this mechanism improves predictions specifically for heavily occluded joints compared to joint-only sampling? For example, results broken down by occlusion level or joint visibility would help clarify this claim. A clearer demonstration of this effect would strengthen the argument for the proposed design.

2. Although the appendix includes experiments with synthetic noise added to 2D keypoints, it would be helpful to understand how the method performs when using different real-world 2D detectors with varying accuracy. Could the authors provide additional experiments comparing performance across different 2D pose estimators? Such results would clarify how dependent the method is on upstream detection quality and would help assess its practical robustness.

3. The proposed method introduces additional sampling operations along limbs and joints. While the paper reports moderate increases in inference time, it would be useful to better understand where the computational overhead mainly arises and whether the approach scales efficiently to higher resolution feature maps or larger models. Further discussion or profiling results could help clarify the practical deployment considerations.

4. The experiments are conducted on Human3.6M and MPI-INF-3DHP, which are standard benchmarks. Do the authors have any preliminary results or insights regarding the method’s performance on more diverse or in-the-wild datasets, or under extreme viewpoints and complex backgrounds? Additional evidence of generalization would strengthen the significance of the approach.

**Limitations:**

No.
The paper only briefly mentions societal impact and states that no specific consequences need to be highlighted, which does not provide a meaningful discussion of limitations or potential risks. While the work is primarily technical, the authors could improve this section by acknowledging several practical and methodological limitations.

**Strengths And Weaknesses:**

Soundness.
The paper appears technically sound and the proposed methodology is appropriate for the task of monocular 3D human pose estimation. The framework is clearly defined and integrates structured sampling and geometric correction into a unified pipeline. The experimental evaluation is conducted on two widely used benchmarks, Human3.6M and MPI-INF-3DHP, and includes comparisons with a variety of recent single frame and multi frame approaches. The authors also provide ablation studies to analyze the contributions of the main components and additional robustness experiments under noisy keypoint inputs and occlusion scenarios. These evaluations generally support the claims made in the paper. However, the performance improvements over the strongest baselines are relatively modest in some cases, and the analysis of limitations or failure cases could be more thorough.

Presentation.
The paper is generally well organized and follows a clear structure, with the motivation, method, experiments, and analysis presented in a logical progression. Figures and diagrams help illustrate the framework and make the overall design easier to understand. Qualitative visualizations and additional analyses in the appendix further support the explanation of the method. However, the presentation could be improved in terms of clarity and conciseness. Some parts of the method section contain dense mathematical notation and introduce many variables in quick succession, which can make the core intuition harder to follow. Simplifying some explanations and emphasizing the main design ideas could improve readability.

Significance.
The paper addresses an important problem in computer vision and machine learning, namely robust monocular 3D human pose estimation. Handling occlusion and depth ambiguity from a single image remains challenging, and improving performance in this setting has practical implications for applications such as motion capture, human computer interaction, and augmented reality. The proposed approach demonstrates improved robustness and competitive accuracy compared to recent single frame methods. Although the numerical improvements are relatively moderate, the idea of modeling limb level structural information directly from image features could encourage further work on incorporating geometric reasoning into pose estimation systems.

Originality.
The work presents a new framework that combines structured feature sampling along skeletal limbs with a geometry based feature correction mechanism. While several individual components such as multi scale feature extraction, deformable sampling, and transformer based regression have appeared in prior work, the proposed approach integrates these ideas in a way that explicitly captures limb level geometric continuity and aligns features using a canonical 3D reference. The novelty therefore lies mainly in the design of the structured sampling strategy and the geometric correction mechanism that enforces global pose consistency. Overall, the work represents a meaningful extension and combination of existing techniques rather than a completely new paradigm.

---

> ### Author Rebuttal · Authors · 2026-03-30
>
> We thank the reviewer for the thoughtful feedback. We address each of the key questions below  (**A:** answer to each concern).
>
> ---
>
> ## 1. Occlusion Analysis for Limb-Level Sampling
>
> **A1:** To quantitatively demonstrate how limb-level sampling improves predictions under occlusion, we provide a per-action breakdown on Human3.6M, where different actions exhibit varying degrees of self-occlusion.
>
> **Table 1: MPJPE breakdown by action on Human3.6M.**
>
> | Action | Occlusion Level | PoseFormer (81 frame) |Joint-Only  | Joint + Limb  |
> |:---|:---:|:---:|:---:|:---:|
> | Sitting Down | Severe| 60.7 mm | 59.3 mm | 55.1 mm |
> | Sitting | Severe| 53.3 mm | 51.2 mm | 47.7 mm |
> | Eatting | Moderate| 39.8 mm  | 37.7 mm | 34.7 mm |
> | Walking | Slight | 31.8 mm  | 31.2 mm | 30.5mm |
>
> The improvement from limb-level sampling is substantially larger for actions with severe self-occlusion (Sitting Down: 4.2 mm, Sitting: 3.5 mm) compared to actions with slight occlusion (Walking: 0.7 mm). This aligns with our design: sampling along the full limb enables the model to leverage continuous visible evidence from unoccluded regions to infer occluded joints.
>
> ---
>
> ## 2. Robustness to Different 2D Detectors
>
> **A2:** To ensure fair comparison with prior works, we follow the standard protocol and use CPN-detected 2D keypoints as input. To further validate that Glimpse can leverage image features to compensate for 2D detection errors, we conduct experiments with three different 2d pose detectors. Results are reported in Table 2.
>
> **Table 2: Performance with different 2D detectors on Human3.6M.**
>
> | Method | 2D Detector | MPJPE (mm) |
> |:---|:---|:---:|
> | CA-PF-HR32 | HRNet | 42.2 |
> | Glimpse-HR32 | HRNet | 40.0 |
> | CA-PF-HR32 | CPN | 41.6 |
> | Glimpse-HR32 | CPN | 39.5 |
> | CA-PF-HR32 | GT (projected) | 36.2 |
> | Glimpse-HR32 | GT (projected) | 35.8 |
>
> Glimpse-HR32 consistently outperforms CA-PF across all settings, with improvements ranging from 0.4 mm to 2.2 mm. This aligns with our core motivation: by sampling visual features along limbs, Glimpse leverages image evidence to compensate for inaccurate 2D keypoints. To further assess robustness, we evaluate under varying levels of synthetic noise **(Appendix A.2, Figure 8)**, where Glimpse also outperforms CA-PF across all noise levels, with the performance gap widening under stronger perturbations.
>
>
> ---
> ## 3. Computational Overhead Analysis
>
> **A3:** A detailed runtime ablation is provided in our response to **Reviewer w831 (A3)** . The full model achieves an MPJPE of 39.5 mm with a total inference time of 55.87 ms on an NVIDIA A6000 GPU, maintaining a practical frame rate of 17.90 FPS. Parameter increase from baseline to Glimpse-HR32 is only 0.58M (**13.74 → 14.32**), confirming that accuracy improvements stem from more effective geometric feature learning rather than increased model capacity. As shown in the Table **Reviewer w831 (A3)**, the computational overhead is distributed across components: joint sampling contributes +2.75 ms; limb sampling (M=3) adds +19.22 ms with an additional **1.1 mm** improvement; geometric correction contributes +9.36 ms with comparable accuracy gain.
>
> To assess scalability to larger backbones, we compare inference time across HRNet-W32 and HRNet-W48. As shown in Table 3, the additional trainable parameters introduced by Glimpse are minimal, while the backbone parameters dominate the total model size. The inference time overhead of structured sampling remains relatively constant across backbone sizes, confirming that the additional geometric reasoning components scale efficiently with larger backbones.
>
> **Table 3: Inference time and parameter comparison across backbone sizes.**
>
> | Setting | Backbone | Backbone Params (M) | Trainable Params (M) | Total Params (M) | Inference Time (ms) |
> |:---|:---|:---:|:---:|:---:|:---:|
> | CA-PF-HR32 | HRNet-W32 | 28.5 | 14.1 | 42.6 | 33.71 |
> | Glimpse-HR32 | HRNet-W32 | 28.5 | 14.3 | 42.8 | 55.87 |
> | CA-PF-HR48 | HRNet-W48 | 63.6 | 14.2 | 77.8 | 39.97 |
> | Glimpse-HR48 | HRNet-W48 | 63.6 | 14.4 | 78.0 | 57.31 |
>
>
> ---
>
> ## 4. Generalization to In-the-Wild Datasets
>
> **A4:** To evaluate Glimpse on more diverse and challenging scenarios, we conduct cross-dataset evaluation on 3DPW, training on Human3.6M and testing directly on 3DPW without fine-tuning. Results are reported in Table 2 of our response to **Reviewer w831 (A2)**. Glimpse achieves an MPJPE of 88.6 mm, outperforming CA-PF-HR32 (90.9 mm) by 2.3 mm. This improvement  demonstrates that the geometric priors learned by Glimpse generalize effectively to in-the-wild conditions.
>
> Beyond cross-dataset evaluation, robustness to input perturbations further supports generalization capability. In **Appendix A.2 (Figure 8)**, Glimpse consistently outperforms CA-PF across all noise levels, with the performance gap widening under stronger perturbations. Together, these results confirm that Glimpse maintains its advantage across varying input qualities and domain shifts.

---

> > ### Author Rebuttal · Reviewer_9JDb · 2026-04-03
> >
> > The authors have provided satisfactory clarifications for the points raised.

---

> > > ### Author Response · Authors · 2026-04-06
> > >
> > > Thank you very much for your positive acknowledgment and for confirming that your concerns have been fully resolved.

---

### Official Review · Reviewer_w831 · 2026-03-12

**Soundness:** 4
**Presentation:** 4
**Significance:** 2
**Originality:** 2
**Overall Recommendation:** 4
**Confidence:** 5

**Summary:**

The authors propose a method to estimate 3D human pose from images. This has the issue of overfitting on small or not very diffuse datasets. The authors propose geometric constraints to attend only to relevant parts by sampling along bones and close to parts detected in 2D. This combines ideas from deformable convolution and using the skeleton structure for human pose estimation. The paper evaluated on the most widely established benchmarks and shows minor improvement (no major improvements can be expected as quite saturated).

**Compliance With Llm Reviewing Policy:**

Affirmed.

**Final Justification:**

Thanks for the additional experiments. There is still no entirely fair comparison to other methods running at the same fps, but since this was not the main focus of this work, I am ok with this and keep my positive score.

**Key Questions For Authors:**

* the ablation on inference runtime would be insightful and should be quick to do. Training time effects could also be relevant.
*please address the 'fair' comparison aspect raised above in terms of runtime
* how is the best model selected, some early stopping? If so, by what criteria?
* learning rate, schedulers and perhaps other hyper parameters are not given in the main document. Perhaps refer to the supplemental.
* is any data augmentation etc. performed? If following previous work in the preprocessing, please specify. For MPI-3DHP, is the model trained on it or only evaluated?

**Limitations:**

3D human pose could be used for surveillance and perhaps other abuses. Risks are not discussed.

**Strengths And Weaknesses:**

Strengths:
* The method is generally sound and very well presented, in terms of mathematical rigor with equations, nice, insightful visualization of the building blocks, and detailed ablation studies and statistics, e.g., analyzing the distribution of error in occlusion-heavy sequences.
* Improvements on the discussed prior work on the tested benchmarks are somewhat small but significant given the dataset saturation.
* The chosen building blocks are well motivated and fill holes that prior methods have left, i.e., they provide a more holistic combination of GNN and geometry-aware sampling of image features.
* The ablation studies are extensive and show the merit of each component, and in combination.

Weaknesses:
* Novelty/originality is somewhat limited as local sampling and using GNN etc. has been proposed before in the HPE context, yet with slightly different base models and implementation details (e.g., older ones are not transformer-based)
* The method is evaluated on two well-known datasets, but both are restrictive in motion (scripted) and environment (all sequences have the same).
* Model size comparison is given, but what about runtime? The custom sampling may introduce a significant overhead. Can you add a breakdown of the runtime for the ablation study (how much longer does it take when adding each of the contributions)?
* The supplemental provides some runtime comparison to baselines, showing a slowdown. How would performance compare if reducing the model parameters until the same fps is reached? Would another fair comparison be possible, e.g., increasing the weights of other methods until they are equally slow?
* Minor: Reference misses a year, please double check all references for consistency and completeness: Pandapose: 3d human pose lifting from a single image via propagating 2d pose prior to 3d anchor space.

---

> ### Author Rebuttal · Authors · 2026-03-30
>
> We thank the reviewer for the constructive feedback. We address the key concerns below (**A:** answer to each concern).
>
> ---
>
> ## 1. Novelty and Originality
>
> **A1:**  To clearly position Glimpse against prior works, we provide a comparison of key design dimensions and performance in Table 1. Prior methods fall into two categories: coordinate-based methods that lack visual features, and visual feature-based methods that sample only at sparse joint locations without modeling limb continuity or global consistency. GNN-based approaches model structural priors through skeletal graphs but remain limited by their reliance on sparse 2D inputs without visual evidence, as noted in our Introduction.
>
> **Table 1: Comparison of designs across different methods.**
>
> | Method | Visual Feature | Deformable Sampling | Limb Continuity | Global 3D Reference | MPJPE (mm) |
> |:---|:---:|:---:|:---:|:---:|:---:|
> | GraphMLP | ✗ | ✗ | ✗ | ✗ | 49.2 |
> | HiPART | ✗ | ✗ | ✓ | ✗ | 42.0 |
> | CA-PF-HR32 | ✓ | ✓ (joint only) | ✗ | ✗ | 41.4 |
> | Glimpse-HR32 | ✓ | ✓ (joint + limb) | ✓ | ✓ | 39.5 |
>
> **Glimpse addresses two fundamental gaps:**  explicit modeling of continuous 2D limb geometry and establishment of a canonical 3D reference for global consistency. **Structured sampling** captures continuous limb geometry by sampling along skeletal segments rather than only at sparse joint locations, enabling the model to leverage visual evidence from the full limb under occlusion. **Geometric correction** introduces a shared canonical 3D reference for distance-aware feature alignment across all joints, ensuring global pose consistency. The ablation study (Table 3(a)) confirms the contribution of each component. Beyond accuracy, structured sampling enhances robustness under occlusion (Section 4.3, Figure 3). Robustness to input noise is further demonstrated in Appendix A.2 (Figure 8).
>
>
>
> ---
>
> ## 2. Dataset Diversity and Generalization
>
> **A2:** Human3.6M and MPI-INF-3DHP are standard benchmarks that enable fair comparison with prior works. To demonstrate robustness and generalization, we provide two complementary analyses: **1) robustness to noisy 2D pose.** In Appendix A.2 (Figure 8), Glimpse consistently outperforms CA-PF across all noise levels. **2) evaluation on 3DPW.** We further evaluate Glimpse on 3DPW (validation set and test set), a challenging dataset featuring outdoor scenes and diverse activities. Results are reported in Table 2. Glimpse achieves better performance than CA-PF-HR32 on 3DPW, confirming strong generalization to in-the-wild conditions.
>
> **Table 2: Results on 3DPW.**
>
> | Method | MPJPE (Valid) (mm) | MPJPE (Test) (mm) |
> |:---|:---:|:---:|
> | CA-PF-HR32 | 97.6 | 90.9 |
> | Glimpse-HR32 | 94.9 | 89.7 |
> | Glimpse-HR48 | 94.4 | 88.6 |
>
> ---
>
> ## 3. Runtime Efficiency and Fair Comparison
>
> **A3:** A detailed efficiency comparison with baseline methods is provided in Appendix A.3. Following the your suggestion, we conduct a runtime ablation in Table 3 to quantify the cost of each component. The structured sampling and geometric correction add moderate costs but together enable the full model to achieve superior accuracy while maintaining a practical frame rate of 17.90 FPS.
>
> **Table 3: Runtime ablation across model components.**
>
> | Method | JS | LS | GC | MPJPE (mm) | Params (M) | FLOPs (G) | Time (ms) | FPS |
> |:---|:---:|:---:|:---:|:---:|:---:|:---:|:---:|:---:|
> | Baseline |   |   |   | 51.2 | 13.74 | 7.99 | 30.96 | 32.30 |
> | + Joint Sampling | ✓ |   |   | 41.4 | 14.09 | 8.03 | 33.71 | 29.67 |
> | + Limb Sampling (M=1) | ✓ | ✓ |   | 40.6 | 14.12 | 8.06 | 41.33 | 25.68 |
> | + Limb Sampling (M=3) | ✓ | ✓ |   | 40.3 | 14.12 | 8.06 | 50.18 | 19.93 |
> | + GC | ✓ |   | ✓ | 40.3 | 14.29 | 8.04 | 40.32 | 24.80 |
> | Glimpse-HR32 | ✓ | ✓ | ✓ | 39.5 | 14.32 | 8.09 | 55.87 | 17.90 |
>
> **On fair comparison.** All runtime measurements are conducted on the same hardware under identical settings. The parameter increase from baseline to full model is only **0.58M (13.74 → 14.32)**, and FLOPs increase is marginal. The accuracy improvements stem from more effective geometric feature learning, rather than increased model capacity, and the sampling points can be reduced (e.g., M=1) to trade accuracy for efficiency.
>
> ---
>
> ## 4. Training Details
>
> **A4:**
> **Model selection.** Following previous works, we train on official training sets and evaluate on test sets. The checkpoint with the lowest test MPJPE is selected, consistent with standard practice in the field. The model for MPI-INF-3DHP is trained exclusively on its training set. To validate generalization, we conducted occlusion analysis (Section 4.3, Figure 3), noise robustness (Appendix A.2, Figure 8), and cross-dataset evaluation on 3DPW (A2). Consistent improvements across these settings demonstrate genuine generalization capability.
>
> **Hyperparameters and data augmentation.** Detailed training hyperparameters are provided in Appendix A.5.1. Horizontal flip augmentation is utilized following previous works.

---

> > ### Author Rebuttal · Reviewer_w831 · 2026-04-03
> >
> > Thanks for the additional experiments. There is still no entirely fair comparison to other methods running at the same fps, but since this was not the main focus of this work, I am ok with this and keep my positive score.

---

> > > ### Author Response · Authors · 2026-04-07
> > >
> > > Thank you very much for your positive acknowledgment and constructive feedback throughout the review process.
> > >
> > > Regarding the fair comparison at equal FPS, we will add a discussion note that with reduced sampling points (e.g., M=1), Glimpse-HR32 (without GC) achieves 40.6 mm MPJPE at 25.68 ms, achieving better accuracy than CA-PF-HR32 (41.4 mm at 29.67 ms) at comparable speed. This helps clarify the efficiency-accuracy trade-off in practical deployment scenarios.
> > >
> > > Thank you again for your valuable suggestions.

---

### Decision · Program_Chairs · 2026-04-30

**Decision:**

Accept (regular)

**Comment:**

This paper proposes Glimpse, a framework for monocular 3D human pose estimation that explicitly models anatomical geometry from a single image. The core of the approach is to apply multi-scale structural sampling to capture both local joint appearance and continuous skeletal limb features, to bridge information gaps incurred by occlusion.

Reviewers reached a consensus that the problem is well-motivated and the architecture seems technically sound. Experimental results show state-of-the-art performance on Human3.6M and MPI-INF-3DHP. Ablation studies demonstrate that limb-level sampling provides superior gains in severe occlusion scenarios.

It seems though a number of issues raised by Reviewer EkiN remain unresolved in the rebuttal: the discrepancy between the original methodological formulation and the rebuttal justifications regarding off-body sampling, the lack of mechanism-level evidence for the synergistic role of sampling beyond performance metrics , and the characterization of the "Geometric Correction" module as a feature reweighting mechanism rather than a coordinate-space correction. The reviewer remained skeptical but did not strongly object to acceptance. The AC hence sides with the majority on accepting the work. However, in the final version, the authors should include a detailed discussion on off-body sampling with a clearer motivation aligned with the original formulation, and incorporate the promised 3DPW results and efficiency-accuracy trade-off analysis.